# Measuring rainfall using microwave links: the influence of temporal sampling

Luuk D. van der Valk[1], Miriam Coenders-Gerrits[1], Rolf W. Hut[1], Aart Overeem[1,2], Bas Walraven[1], and Remko Uijlenhoet[1]

[1]Department of Water Management, Delft University of Technology, Delft, the Netherlands
[2]Royal Netherlands Meteorological Institute, De Bilt, Netherlands

**Correspondence:** Luuk van der Valk (l.d.vandervalk@tudelft.nl)

**Abstract.** Terrestrial microwave links are increasingly being used to estimate path-averaged precipitation by determining the attenuation caused by rainfall along the link path, mostly with commercial microwave links from cellular telecommunication networks. However, the temporal resolution of and method to derive these rainfall estimates is often determined by the temporal sampling strategy that is employed by the mobile network operators. Currently, the links are most often sampled at a temporal resolution of 15 minutes with a recording of the minimum and maximum values, while more recently also a form of instantaneous sampling with possible intervals up to 1 s has been set up. For rainfall research purposes, often high temporal resolutions in combination with averaged values are preferred. However, it is uncertain how these various temporal sampling strategies affect the estimated rainfall intensity. Here we aim to understand how temporal sampling strategies affect the measured rainfall intensities using microwave links. To do so, we use data from three collocated microwave links, two 38 GHz and one 26 GHz, sampled at 20 Hz and covering a 2.2 km path over the city of Wageningen, the Netherlands. We aggregate the microwave link power levels to multiple time intervals (1 s to 60 min) and use a mean, instantaneous, and minimum and maximum value to characterize the signal. Based on the aggregated data, we compute rainfall intensities and compare these with 20 Hz rainfall estimates, such that we isolate errors and uncertainties caused by the sampling strategies from instrumental effects, such as different biases between instruments and representativeness errors. In general, our results show that for all sampling strategies an increase in sampling time interval reduces the performance of the rainfall estimates, which especially holds for the instantaneous sampling strategy. Even the mean sampling strategy, which generally performs best of all strategies, is sensitive to this reduction in temporal resolution and could lead to significant underestimations. This sensitivity of the mean sampling to the temporal resolution seems to be largely affected by the non-linear relation between attenuation and rainfall. The min-max sampling strategy is mostly prone to minor underestimations or large overestimations of the path-averaged rainfall intensities. Moreover, our results, including a comparison with theoretical events, show that the attenuation due to wet antennas not only affects the comparison between the rainfall estimates obtained with a microwave link and another reference instrument, but also has a significant influence on the performance of the rainfall retrieval algorithm, especially for devices with relatively long duration of the wet-antenna attenuation combined with the longer sampling time intervals. Overall, this study demonstrates the effect a selected sampling strategy can have on rainfall intensity estimates using (commercial) microwave links.

## 1  Introduction

Accurate rainfall measurements are essential in many fields of application. For example, water resources management uses rainfall measurements for flood forecasting (e.g., Maggioni et al., 2018), a major part of global agriculture is dependent on rain (Molden, 2013) and urban runoff estimates are highly dependent on rainfall estimates (e.g., Berne et al., 2004; Cristiano et al., 2017; Niemczynowicz, 1988). Overall, these practices would benefit from increasing the spatial and temporal resolution of rainfall measurements.

Currently, dedicated rainfall measurement techniques have some important drawbacks. Ground-based point measurement devices, such as rain gauges or disdrometers, are often able to capture the temporal dynamics of precipitation, but do not represent the spatial character of precipitation (e.g., Berne et al., 2004; Sun et al., 2018). Moreover, the placement of the devices can decrease the measurement performance, for example through the wind causing an undercatch (e.g., Pollock et al., 2018; Raupach and Berne, 2015). Weather radars do provide the desired spatial rainfall information combined with a sufficient temporal resolution. However, radars measure higher up in the atmosphere and indirectly retrieves rainfall introducing uncertainty about the actual amount of precipitation near the surface (Berne and Krajewski, 2013). Additionally, both methods are not available on a global scale, due to costs and maintenance. On a global scale, including the oceans, satellites provide rainfall information, but for hydrometeorological applications these can come at a too low spatial and temporal resolution combined with too high uncertainty and bias, partly dependent on, for example, terrain complexity, aridity and season (Maggioni et al., 2018; Rios Gaona et al., 2017). Additionally, satellite rainfall products, and especially merged products, have a relatively long latency (e.g., IMERG has about a 4 hour latency for the earliest run; NASA, 2024).

Another source of spatial rainfall estimates could come from telecommunication networks, a so-called opportunistic sensing technique. These networks consist of commercial microwave links (CMLs), the rain-induced attenuation of the electromagnetic signal of which can be used to compute rainfall intensities (e.g., Chwala and Kunstmann, 2019; Messer et al., 2006; Leijnse et al., 2007a; Uijlenhoet et al., 2018). These CMLs are near-surface radio connections used in cellular telecommunication networks. Thus, as a major advantage, the infrastructure required to spatially measure rainfall with these CMLs already exists. Furthermore, the rainfall estimates obtained with a single link are representative for the entire path, overcoming the drawbacks of point measurements. As shown by de Vos et al. (2018), CMLs are especially useful when considering spatial aggregation scales that are too large to cover entirely with point measurements. Also, for many applications spatial rainfall estimates on scales in the order of a couple kilometres are more relevant than point measurements.

Moreover, rainfall measurements by a CML network can be beneficial in combination with other rainfall measurements and are increasingly being used in hydrometeorological applications. For example, Brauer et al. (2016) used CML rainfall estimates as input in a rainfall-runoff model for lowland catchments and showed, in general, that these rainfall estimates are very suitable as input for hydrological applications. van het Schip et al. (2017) showed the complementary potential of CML and satellite data by determining wet and dry periods using the satellite data, while Hoedjes et al. (2014) proposed to use this for a conceptual flash flood early warning system in Kenya. Fencl et al. (2013) and Pastorek et al. (2023) used CMLs as input data for an urban drainage model and demonstrated the benefits of the relatively high spatial resolution on these

models. Also, Imhoff et al. (2020) showed that nowcasting rainfall events could be performed using CML networks, with good results when compared to weather radar precipitation estimates and nowcasts. This is especially promising for regions without weather radars. Moreover, the number of CMLs operating worldwide in the 6-56 GHz range, which are most useful for rainfall estimation, is expected to grow from 4.6 million in 2021 to 6 million in 2027 (ABI research, 2021). Overall, this shows the potential of using CML networks for rainfall measurements in many societally relevant hydrometeorological applications.

In general, the attenuation of a microwave link signal can be converted to rainfall intensity using (Atlas and Ulbrich, 1977; Olsen et al., 1978):

$$R = ak^b, \tag{1}$$

where $R$ is the rainfall intensity (mm h$^{-1}$), $k$ the specific attenuation (dB km$^{-1}$) and $a$ and $b$ are coefficients depending on both signal characteristics (e.g., frequency and polarization) and precipitation characteristics (e.g., drop size distribution) (Jameson, 1991). For frequencies typically applied in CML networks, $b$ is close to 1, so that signal attenuation and rainfall intensity are nearly proportional. Messer et al. (2006) and Leijnse et al. (2007a) showed that the commercially-employed microwave links could also be used to measure rainfall intensities, in Israel and the Netherlands, respectively. Since then, studies have been performed in Europe (Czech Republic, France, Germany, Italy, Luxembourg, Sweden and Switzerland), Africa (Burkina Faso, Kenya and Nigeria), South America (Brazil), Asia (Lebanon, Pakistan and Sri Lanka) and Oceania (Australia and Papua New Guinea) (see Chwala and Kunstmann, 2019, for a partial overview). Several open-source packages exist for rainfall retrieval (and mapping) from CML data. One of these is RAINLINK (Overeem et al., 2016b). Chwala et al. (2016) have developed an algorithm to extract real-time data from CML networks and based on this data determined spatial rainfall estimates (Graf et al., 2020). Habi and Messer (2021) used a recurrent neural network to determine rainfall intensities using CML network data.

However, using these CML networks to estimate precipitation, the temporal resolution of these estimates is often bound to the temporal sampling strategy employed by the mobile network operator, which solely uses the information on the link signal to assure the functioning of the network. Moreover, not all mobile network operators store the same variables describing the link signal in their network management system. Minimum and maximum values (and occasionally mean and/or instantaneous values) are most commonly measured with a temporal resolution of 15 minutes. Additionally, the method developed by Chwala et al. (2016) allows to actively select any instantaneous sampling method up to 1 s intervals for these networks and is specifically designed to estimate rainfall intensities, in contrast to using the data with the sampling strategy chosen by the network operators. For research purposes, data with a high temporal resolution is often preferred, so that the temporal sampling resolution is higher than the dominant timescales of rainfall. Previously, Leijnse et al. (2008) showed that different sampling strategies together with nonlinearities in the $R - k$ relationship are the dominating errors when estimating rainfall. For this analysis, Leijnse et al. (2008) used microwave link simulations based on radar data with a 16 s interval and only studied the effect for 15 minute sampling strategies. Yet, it is uncertain how larger variations in sampling strategies affect the computed amount and intensity of rainfall together with the use of actual microwave link data.

Here, we aim to understand how temporal sampling strategies affect the measured rainfall intensities. To do so, we compare 20 Hz rainfall estimates obtained with 38 and 26 GHz microwave links with rainfall estimates computed with various temporal sampling strategies. One of these links has formerly been employed in an operational CML network. Ultimately, this allows us to isolate errors and uncertainties caused by the sampling strategy from instrumental effects. As a consequence of only having one sampling strategy set by the mobile network operators, these errors and uncertainties usually cannot be separated due to the comparison of different instruments, which represent different measurement volumes combined with different measurement uncertainties. Thus, using this method, we are able to estimate the actual errors and uncertainties due to the sampling strategy. We do so for commonly selected sampling strategies (e.g., a minimum and maximum intensity per 15 minutes) as well as less or never selected sampling strategies, which allows to systematically illustrate the sensitivity of the measured rainfall intensities to the various sampling strategies. Overall, this could help to estimate the effects of the strategies set by mobile network operators or help to choose an optimal strategy when estimating rainfall intensities using CMLs.

## 2  Methods

### 2.1  Instrumentation

In this study, we use data published online from van Leth et al. (2018a), who reported on a measurement campaign (see van Leth et al., 2018b) using three microwave links along a 2.2 km path over the city Wageningen, the Netherlands (Fig. 1). In this section, we describe the essential information required for understanding our analysis. For a more elaborate description, we refer the reader to van Leth et al. (2018a).

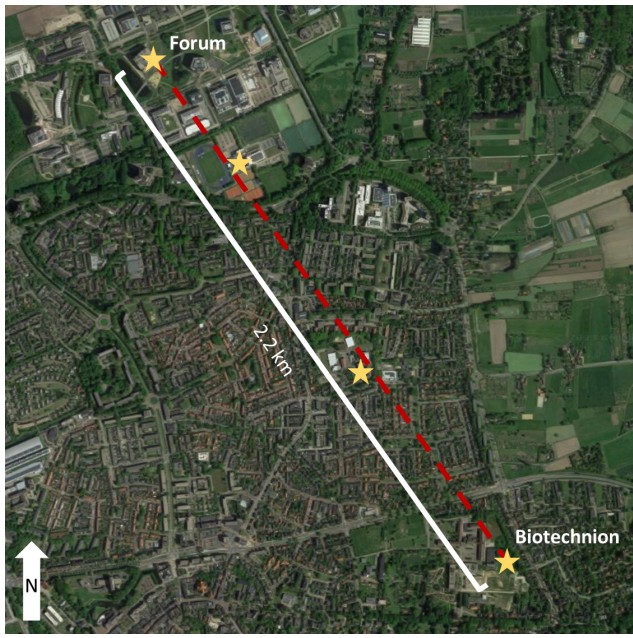

**Figure 1.** Map of Wageningen with the path of the links in red. The receiving antennas are at the end labelled "Forum"; the transmitting antennas are positioned at the end labelled "Biotechnion". The yellow stars indicate the position of disdrometers along the path. At each disdrometer location, one disdrometer is located, except at "Forum", which has two disdrometers placed next to eachother. Figure redrawn after van Leth et al. (2018a) (©Google Maps).

We use data from a Nokia Flexihopper transmitting at 38.2 GHz with a bandwidth of 0.9 MHz, which was formerly part of the cellular communication network operated by T-Mobile NL, and two links built by Rutherford Appleton Laboratories
(RAL) transmitting at 26.0 and 38.0 GHz, with a receiving bandwidth of 4 KHz and transmitting bandwidth of much less than 1 KHz. All the microwave link signals are sampled at 20 Hz using logarithmic detectors. The employed frequencies for the Nokia and RAL 38 GHz links are close, hence exhibit similar electromagnetic characteristics, but do not interfere with each other. However, these devices were found to give a different response, likely due to the internal hardware in the Nokia link being designed differently, reducing the high-frequency fluctuations in the signal, while the RAL link has a different antenna
cover than the Nokia link, which affects the distribution of water remnants on the cover (see van Leth et al., 2018a). On the RAL cover water droplets form once it gets wet, which induces a more significant attenuation of signal intensity than the water film that forms on the Nokia cover after getting wet. Eventually, these wet antennas cause an additional attenuation of the signal, which causes an overestimation of rainfall intensities following the $R - k$ relation if not accounted for. The RAL 26 GHz link is less prone to wet-antenna attenuation than the RAL 38 GHz link.
In this paper we only consider horizontal polarization, since for both the Nokia and the RAL 26 GHz links only this polarization is available. For the RAL 38 GHz link, both horizontal and vertical polarized data are available, but we only show results for horizontal polarization, due to the insignificant differences between these polarizations in our results and to make a fair comparison with the other devices. Results for the vertically polarized signal are included in the Supplementary materials.

In this study, we use the disdrometer data to distinguish wet and dry periods and filter out snow, hail, graupel and mixed precipitation events when comparing the rainfall intensities between the 20 Hz data and other temporal sampling steps. Note that it is not our aim to compare the rainfall intensities from the disdrometers with the rainfall estimates from the microwave links, because we aim to understand the influence of the temporal sampling strategies on estimated rainfall intensities. Along the 2.2 km path, five OTT Parsivel laser disdrometers were installed in order to compute path-averaged rainfall estimates, of which we also obtained the post-processed rainfall intensities and precipitation types, i.e., to remove non-liquid precipitation, from van Leth et al. (2018a). The disdrometers measured raindrop size distributions every 30 s and were post-processed using the method of Raupach and Berne (2015), which corrects for instrumental biases. van Leth et al. (2018a) compared rainfall estimates of the microwave links and the disdrometers and show large additive and multiplicative biases for all instruments. The RAL links exhibit an additive bias around 2.2 mm h$^{-1}$, while the Nokia link has an additive bias of 0.6 mm h$^{-1}$. The multiplicative bias for all instruments ranges between 1.5 and 1.7.

The microwave links were operational over a period from 22 August 2014 to 8 January 2016, but not all disdrometers were operational during the entire period. Therefore, we use the data from the start of April 2015 to the end of December 2015, so that we have fully operational instruments except for a power outage from 7 to 25 August 2015. By using all the disdrometers along the path (instead of a single disdrometer), we decrease the uncertainty in the wet-dry classification and incorrect removal of other precipitation types, such that errors and uncertainties arising in our results are most likely to originate from the microwave link rainfall estimates.

To identify relevant precipitation climatologies for our study area, we examine the disdrometer data. Note that the data presented pertains to a 2.2 km path and not necessarily reflects exactly the same climatology as for the whole country or region. This reveals that the average duration of a precipitation event longer than 5 minutes in our dataset is roughly 30 minutes, with a median duration of 12 minutes (Fig. 2), demonstrating a positively skewed distribution with many short rainfall events and relatively few longer rainfall events. In this figure two contiguous rainy periods, which are separated by a single 30-second dry time interval, are counted as separate events and not combined into a single event. To refrain ourselves from making any assumptions about when a rainfall event is continuous or not, we decided to use a single timestep in the available disdrometer data as threshold. The rainfall intensities show in general higher peak intensities during summer than winter. Due to the power outage between 7 and 25 August, the data for that month can be less reliable.

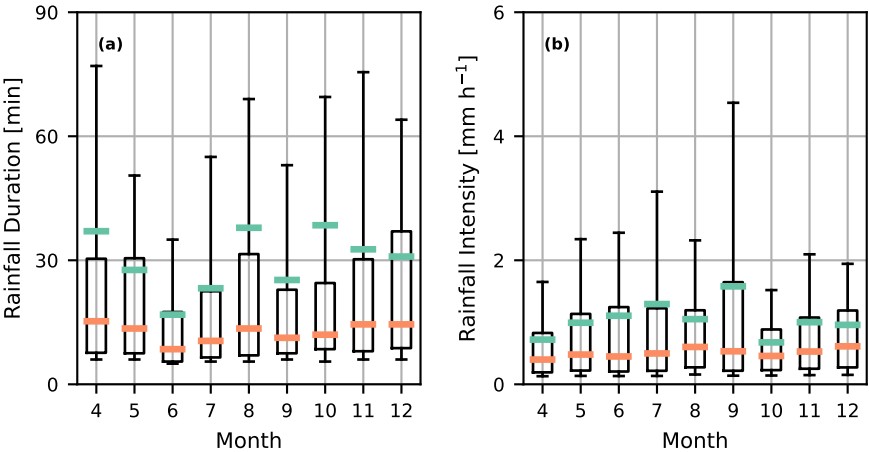

**Figure 2.** Boxplots describing the duration of rainfall events per month in 2015 (a) and all rainfall intensities per month in 2015 (b). In the boxplots, orange is the median and green is the mean duration or intensity. The box edges represent the first and third quartiles, and the whiskers are the 10[th] and 90[th] percentiles. The values are computed using disdrometers installed along the 2.2 km microwave link path. In the duration statistics, events shorter than 5 minutes are excluded. For the rainfall intensities, all rainfall intensities larger than 0.1 mm h[-1] are used.

## 2.2 Rainfall retrieval from microwave links

To identify the influence of temporal sampling on rainfall estimates, we compare rainfall intensities obtained by using various sampling strategies with high-resolution rainfall estimates. We do this for the three devices sampling a mean, instantaneous, and minimum and maximum ("min-max strategy") value per time interval, which mimic various temporal sampling strategies by network operators. As sampling time interval we use 1 s, 30 s, 1 min, 5 min, 15 min, 30 min and 60 min. As reference data, we use rainfall intensities obtained using the 20 Hz sampling of the same device, in order to be able to solely study the influence of the sampling strategies on the measured rainfall intensities.

We use a relatively straight-forward method to compute the rainfall intensities, to limit the effect of many parameters in the algorithm. The rainfall measurements are done in a similar fashion as van Leth et al. (2018a) combined with the method of Overeem et al. (2016b) for the min-max sampling strategy. This means no corrections on the rainfall estimates, for example for wet-antenna attenuation. The retrieval algorithm is as follows:

1. Aggregate microwave link power levels to the mean, minimum, maximum and instantaneous value of time intervals: 1 s, 30 s, 1 min, 5 min, 15 min, 30 min and 60 min. For instantaneous sampling strategies, we chose to use the last value per time interval to estimate the rainfall intensity for the entire time interval, similar to data provided by mobile network operators.

2. For each sampling strategy, including the original 20 Hz data, determine the baseline signal power level for each interval by selecting the median power level during all dry periods in the preceding 24 h (Overeem et al., 2016b). In the next

step, this baseline power level is used to determine the rain-induced attenuation. For the min-max sampling strategy, the baseline power level is obtained by averaging the maximum and minimum received power levels, assuming a symmetrical distribution of these values, and subsequently computing the median of the preceding 24 h. Of the previous 24 h, at least 1 h should be dry in order to determine a baseline. If all of these time intervals indicate rain, a baseline cannot be determined, and as such rain intensities cannot be determined. The selection of dry periods is based on disdrometer data. This method of baseline determination is based on Overeem et al. (2011).

3. Based on the power and baseline levels, compute the specific attenuation of the signal $k$ (dB km$^{-1}$),

$$k = \frac{P_{\mathrm{ref}} - P}{L}, \tag{2}$$

in which $P_{\mathrm{ref}}$ is the baseline power level, $P$ the received power levels (both in dBm) and $L$ is the path length (km). For the min-max sampling, the specific attenuation is calculated for both the minimum and maximum attenuation using the same baseline for both.

4. Convert the specific attenuation to rainfall intensity, using Equation 1. The values for the parameters $a$ and $b$ are the same as applied by van Leth et al. (2018a) (Table 1). They obtained these values from non-linear least-squares fits of disdrometer-derived rainfall intensities and specific attenuations at the frequencies employed by the microwave links combined with scattering computations. For the min-max sampling, the rainfall intensities using the minimum and maximum attenuation are calculated separately and are combined into one rainfall intensity in step 6. These values differ from the recommendations by ITU-R (2005).

**Table 1.** Values for the $a$ (mm h$^{-1}$ dB$^{-b}$ km$^{b}$) and $b$ (–) parameters in Eq. 1 specifically derived for the dataset from van Leth et al. (2018a) and general recommendations by the ITU-R (2005) for 38 and 26 GHz horizontally polarized signals.

|  | $a_{38\mathrm{GHz}}$ | $b_{38\mathrm{GHz}}$ | $a_{26\mathrm{GHz}}$ | $b_{26\mathrm{GHz}}$ |
|---|---|---|---|---|
| van Leth et al. (2018a) | 3.83 | 1.05 | 7.70 | 0.93 |
| ITU-R (2005) | 2.82 | 1.13 | 5.92 | 1.01 |

5. Set the computed rainfall intensities during dry periods (based on disdrometer data) to zero. Since we treat the data as if they originate from a CML network, this implies that if it rains part of a time interval, the whole interval is seen as wet. Also, for the comparison of all rainfall intensities, we remove snow, hail, graupel and mixed rain events from the data based on the disdrometer data, since these have a different effect on the signal in comparison to rain.

6. For the min-max sampling strategy, combine the rainfall intensities obtained using the minimum attenuation and maximum attenuation following

$$R = \alpha R_{\mathrm{max}} + (1 - \alpha) R_{\mathrm{min}}, \tag{3}$$

in which $R$ is the estimated mean (based on the minimum and maximum rainfall intensities) rainfall intensity, $R_{\mathrm{max}}$ and $R_{\mathrm{min}}$ the rainfall intensities derived from the maximum and minimum attenuations (mm h⁻¹), and $\alpha$ is the parameter determining the contribution of the minimum and maximum received power levels. In this study, we use an optimized $\alpha$ and a non-optimized $\alpha$ of 0.33, as determined for an operational microwave link network in the Netherlands by Overeem et al. (2011). The optimization of $\alpha$ is done for each device and sampling strategy by comparing the rainfall intensities obtained through min-max sampling per time interval with the 20 Hz rainfall intensities and selecting the $\alpha$ for which the root mean square error (RMSE) is lowest in combination with an absolute mean bias error (MBE) smaller than 0.02 mm h⁻¹. Overeem et al. (2011) used the residual standard deviation for this, which is the same as the RMSE if the MBE is equal to zero. For this computation, values of $\alpha$ ranging between 0 and 1 with intervals of 0.001 are used. The 20 Hz intensities are averaged to the same time interval as the min-max sampled intensity. The RMSE is computed as

$$\mathrm{RMSE} = \sqrt{\frac{\sum(R_{\mathrm{obs}} - R_{\mathrm{20Hz}})^2}{n}}, \tag{4}$$

in which $R_{\mathrm{obs}}$ is the observed rainfall intensity [mm h⁻¹] with the sampling strategy, $R_{\mathrm{20Hz}}$ the reference rainfall intensity with the 20 Hz sampling strategy and $n$ the number of observations. MBE is computed as

$$\mathrm{MBE} = \frac{\sum(R_{\mathrm{obs}} - R_{\mathrm{20Hz}})}{n}. \tag{5}$$

Note that, unless specifically mentioned, when we refer to min-max sampling strategies in the text and figures, we refer to the optimized version of this sampling strategy. This way, we prevent introducing an additional source of error and uncertainty into the study.

7. To compare the obtained rainfall intensities for the 20 Hz sampling with the other time intervals, we apply a linear regression in which the 20 Hz estimates are the independent variable and the estimates from the other sampling strategies are the dependent variable. To do so, we use again the 20 Hz rainfall intensities averaged to the various time intervals. We compute the MBE, RMSE, r², representing the fraction of explained variance, and the slope of the fit (without intercept). r² is computed as

$$r^2 = 1 - \frac{\sum(R_{obs} - R_{20Hz})^2}{\sum(R_{obs} - \overline{R_{obs}})^2}, \tag{6}$$

in which $\overline{R_{obs}}$ the average observed rainfall intensity.

Note that averaging the 20 Hz rainfall estimates is not the same as the mean sampling, since the averaging occurs in a different step during the rainfall intensity computation and the $R - k$ relation is nonlinear (Eq. 1). This causes the rain intensities obtained by using averaged attenuation to be different compared to the averaged rainfall intensities. For example, if a rain event takes place exactly for 30 seconds in an hour with an attenuation of 10 dB km⁻¹ (i.e. on average

0.083 dB km⁻¹ for an hour), this would result in a decrease of the rainfall for the 60-minute estimate in comparison to the estimate based on the 30-second time interval:

$$\frac{\hat{R}}{\overline{R}} = \frac{ak_{60min}^b}{ak_{30s}^b/120} = \frac{(10/120)^{1.05}}{10^{1.05}/120} = 0.79,$$ (7)

in which $\overline{R}$ is the reference averaged rainfall intensity, $\hat{R}$ the estimated averaged rainfall intensity and the $b$ value is used for a 38 GHz horizontally polarized device (Table 1). At 26 GHz link, for which $b$ is 0.93, this equation would result in 1.40. Other differences could for example exist between baseline power levels, due to not setting the exact same part of the time series to dry as a consequence of time intervals during which it rains only partially (step 5).

## 2.3   Comparison with theoretical events

To examine the influence of precipitation climatology on our results, we compare the obtained relationships between device and sampling strategies to theoretical rain events. We prescribe three types of rain events: 1) a rain event with a constant intensity of 5 mm h⁻¹, 2) a high-intensity event with a maximum intensity of 40 mm h⁻¹ and declining following an exponential function and 3) a low-intensity event, which we compute using three sinusoidal functions (Fig. 3). All events take two hours. The latter two events resemble the two real rain events that are studied in Sect. 3.1. First, we recalculate these events to 20 Hz power levels using Eqs. 1 and 2, subsequently we add normally distributed noise with a device-specific standard deviation to these power levels. This device-specific standard deviation is computed by averaging hourly standard deviations for all dry hours in the dataset. For the Nokia link, this results in a standard deviation of 0.15 dB and for both RAL links in 0.20 dB. Using this method, we obtain a similar dataset as van Leth et al. (2018a), so that we can apply the algorithm as described above. We apply the normally distributed noise for 100 different random states to remove the influence of a single random state on the comparison. After averaging the statistical metrics of these states, we compare these metrics for the theoretical events with the dataset from van Leth et al. (2018a) in order to attribute which part of the uncertainties originates from the algorithm and which part from the actual rainfall variations. Additionally, we study the influence of the starting times of the theoretical rain events on the statistical metrics. To do so, we create 30 theoretical events each with a shift in starting time compared to the original theoretical events. These shifts are uniformly distributed between 2 and 60 minutes (i.e., a 2-min interval). This allows us to estimate the robustness of the sampling strategies against (possibly unfavourable) starting times of rain events.

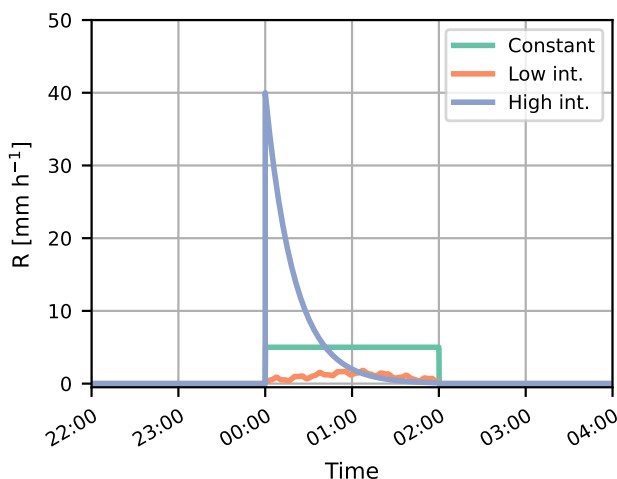

**Figure 3.** Three theoretical rainfall events for estimating the sole influence of the sampling strategies. The constant intensity is 5 mm h$^{-1}$. The low-intensity event consists of three sinusoidal waves: $R = 1 - 0.5\cos(\frac{2\pi \cdot t}{2}) - 0.25\cos(\frac{2\pi \cdot t}{0.25}) + 0.1\sin(\frac{2\pi \cdot t}{0.1})$. The high-intensity event follows the function: $R = 40\exp(-3t)$. In these $t$ is the time (h) since the start of the event.

## 3 Results

### 3.1 Influence of temporal sampling on rainfall estimates for two individual events

We consider two individual precipitation events to illustrate the influence of temporal sampling on estimating rainfall. Figure 4, a low-intensity event, and Fig. 5, a high-intensity event, include the 1 s, 5 min and 60 min time intervals, to demonstrate the influence of the sampling methods on individual events (see Figs. S5 and S6 for the conventional 1-minute and 15-minute time intervals). The analysis in Sect. 3.3 also includes the other time intervals.

When comparing the signals of all devices during the two precipitation events, the largest differences between the Nokia 250 (a-h) and the RAL links (i-p and q-x) are the reduced wet-antenna attenuation and fluctuations in the signal for the Nokia link. For example, considering the wet-antenna attenuation, the tail after the high-intensity event, around 11:00, shows 1-2 dB larger attenuation for the RAL links compared to the Nokia link (Fig. 5a & i). These differences in wet-antenna attenuation for the Nokia and RAL 38 GHz links are caused by different behaviour of rain remnants on the antenna covers, in the form of a water film and droplets, respectively (Fig. 14 in van Leth et al. (2018a)). In general, this causes the wet-antenna attenuation for the 255 RAL link to be roughly 2 dB higher than for the Nokia link shortly after a rain event. For this specific setup, this would result in roughly a 4 mm h$^{-1}$ increased rainfall intensity. The differences in the fluctuations are clearest in the 20 Hz signal for all devices (e.g., Figs. 4a, i and q). These high-frequency fluctuations in the signal are roughly reduced by 0.5 dB, which is likely caused by the different internal electronics in the Nokia link. Yet, this only seems to affect the difference between minimum and maximum intensity values (Figs. 4d, l and t), but not the computed rainfall intensity using this sampling strategy (Figs. 4h,

p and x). Compared to the RAL 38 GHz link, the RAL 26 GHz link shows slightly more fluctuations in the estimated rainfall intensities for the min-max sampling strategies, which could possibly be caused by the different exponent in the R-k relation.

During the event on 24 November 2015, the baseline changes slightly during the precipitation event. In the preceding 24 hours to this event, the signal intensity fluctuates combined with precipitation events. As a result, during the 12 hours of the event the baseline algorithm computes slight changes in power level, as the initial baseline power levels are not based on the same power levels as the final baseline power levels. We expect that this occurs throughout the entire dataset and not only for this specific event.

When comparing the sampling strategies, the instantaneous sampling strategy shows the largest sensitivity of the estimated rainfall intensities to the sampling time interval. Especially for the longest time intervals with the RAL 26 GHz and 38 GHz links, the intensities derived with instantaneous sampling show major fluctuations, which are not necessarily fully representative for the entire interval. This mostly holds for the event on 24 November, as the signal-to-noise ratio of the event on 21 June is relatively large. The min-max sampling strategy shows a large sensitivity to the extreme values, either caused by a rainfall event or an outlier in signal intensity. This especially holds for the maximum attenuation, as the minimum attenuation is bounded at zero (i.e., no attenuation), while the maximum attenuation could theoretically attain very large values (e.g., more than 20 dB in Fig. 5d and l around 10:00). Overall, this seems to cause that the min-max sampling strategy is mostly prone to overestimations, especially for the higher rainfall intensity event on 21 June. In general, the mean sampling seems to represent the 20 Hz rainfall intensities best, though with some minor differences between the sampled time intervals.

Generally, the instantaneous and min-max sampling strategies seem to be more prone to errors in retrieving high rainfall intensities. For the event on 21 June (Fig. 5), the mean sampling strategy results, as expected, in decent estimates of the evolution of the rainfall event on average, both in timing and intensities (though the peak rainfall intensities are obviously averaged out). The instantaneous sampling strategy seems to be most sensitive to an increase in the length of the time interval, because its performance depends on the representativeness of a single measurement for the whole time interval. The timing and intensity of the estimated peak rainfall intensity for the shortest time intervals are often relatively good, due to the large signal intensity fluctuations as a consequence of variations in rainfall intensity in comparison to the instrument noise, while for longer time intervals the sensitivity of the performance to the representativeness of a single measurement for the whole time interval heavily increases (e.g., the 5-minute instantaneous sampling strategy in Fig. 5 captures the peak intensity at 12:00 better than the mean and min-max sampling strategies). For the min-max sampling strategy, the timing of the peak intensity is generally well-captured, but the estimated peak rainfall intensity can be inaccurate. Additionally, for this specific case, this method strongly overestimates the rainfall sum for the 60-minute interval, due to the peak taking place around the full hour, so that two subsequent intervals cover this peak.

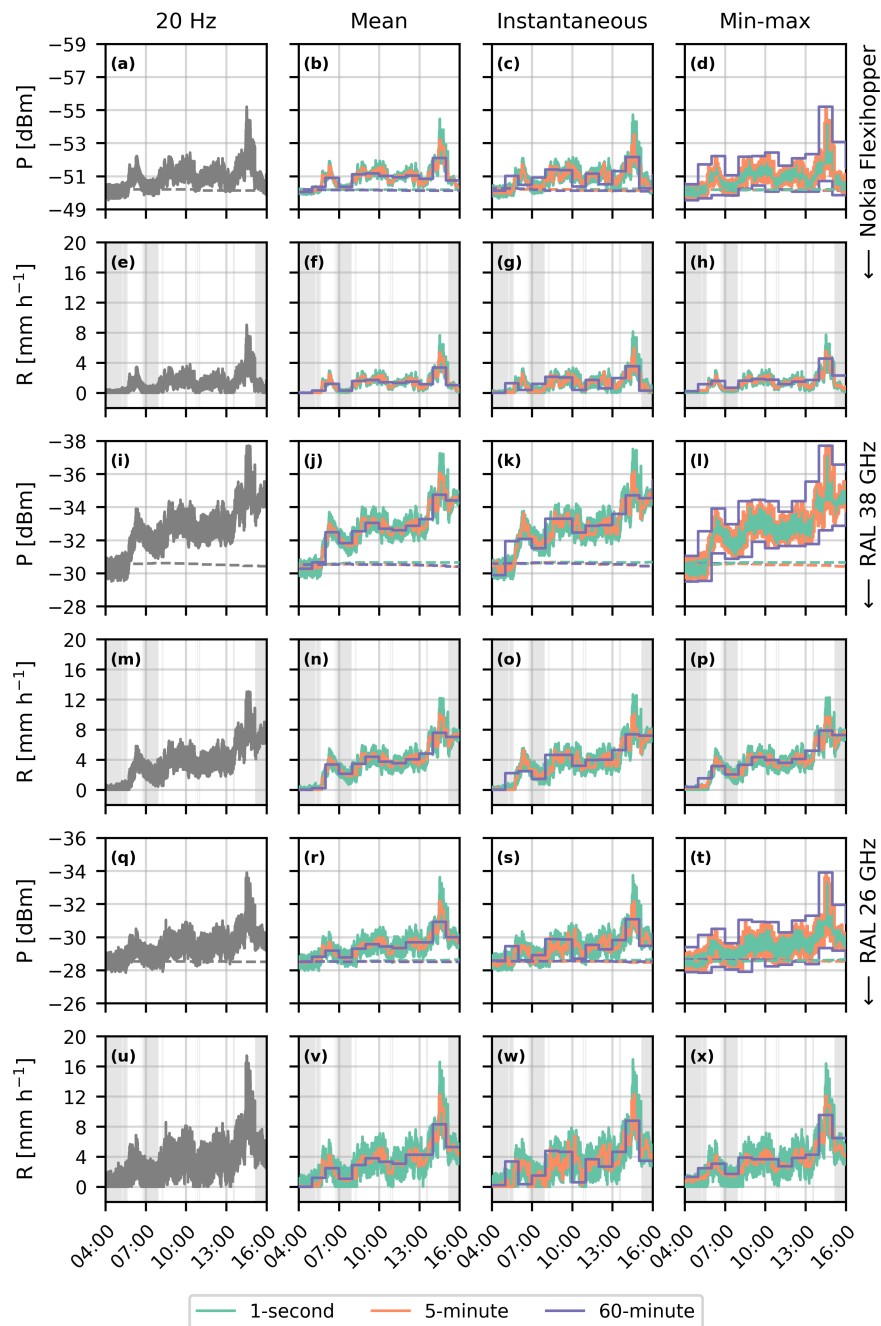

**Figure 4.** Comparison of received (solid) and baseline (dashed) power levels (a-d, i-l, q-t) and retrieved rainfall intensities (e-h, m-p, u-x) during a low-intensity precipitation event on 24 November 2015 obtained with the Nokia (a-h), RAL 38 GHz (i-p) and RAL 26 GHz (q-x) microwave links for all sampled variables and the 1-second, 5-minute and 60-minute time intervals. Grey areas indicate dry periods based on disdrometer data.

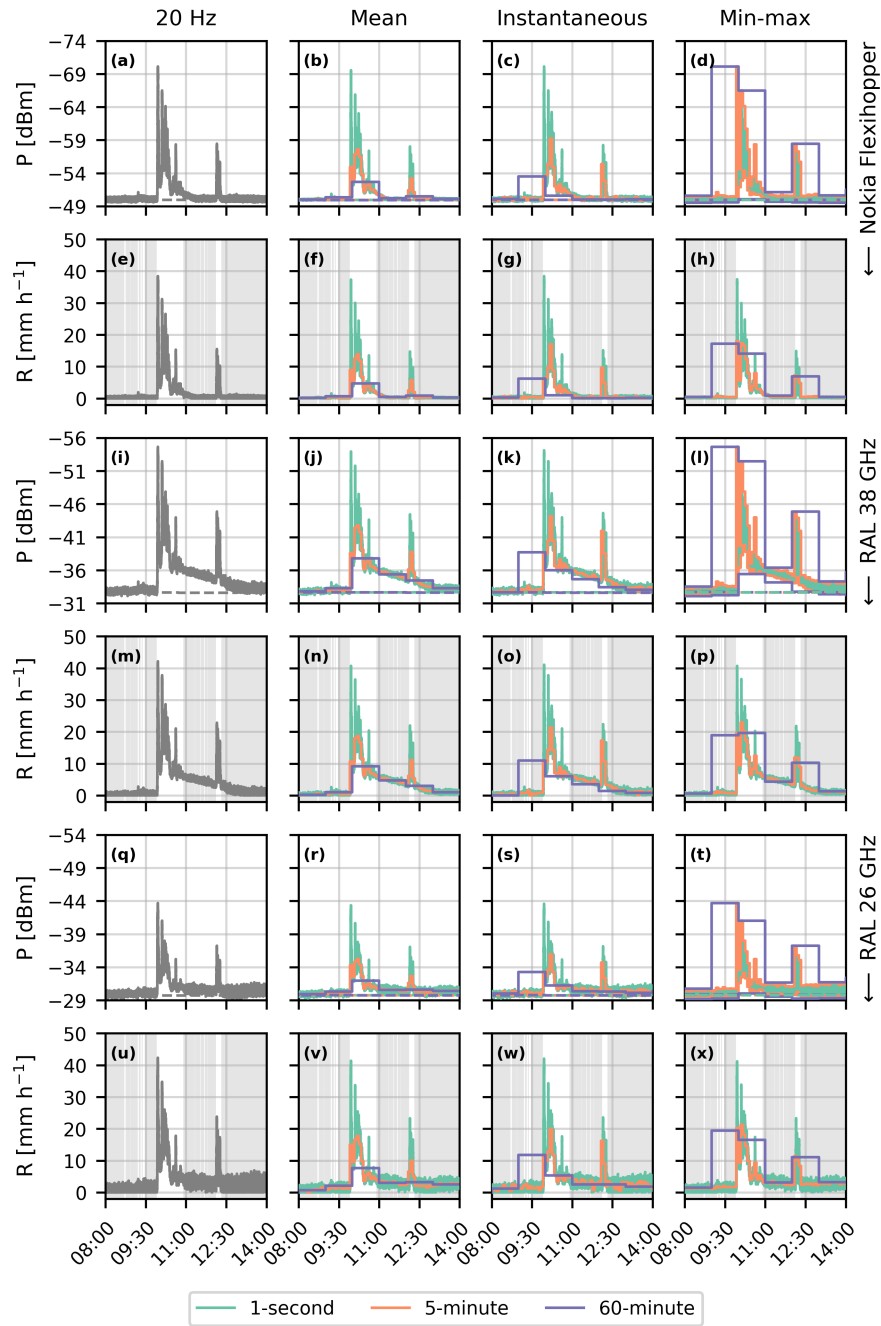

**Figure 5.** Comparison of received (solid) and baseline (dashed) power levels (a-d, i-l, q-t) and retrieved rainfall intensities (e-h, m-p, u-x) during a high-intensity precipitation event on 21 June 2015 obtained with the Nokia (a-h), RAL 38 GHz (i-p) and RAL 26 GHz (q-x) microwave links for all sampled variables and the 1-second, 5-minute and 60-minute time intervals. Grey areas indicate dry periods based on disdrometer data.

## 3.2 Theoretical events

Before analysing all rain events in the whole Wageningen dataset, we examine the theoretical events. This allows us to determine which part of the uncertainties originates from the rainfall retrieval algorithm and which from the instruments. We compare the statistical metrics of three theoretical rain events to which we added three levels of noise, so that these resemble the microwave link signals of the three devices (Fig. 6). Additionally, we compare these with the statistical metrics of the same theoretical events, but without the added noise, which resemble a 38 and a 26 GHz horizontally polarized link, in order to estimate the influence noise has on our comparison (Fig. 7).

The theoretical events demonstrate the sensitivity of instantaneous sampling to individual events, for both the noisy and noiseless signals, especially for longer time intervals. This sampling shows an erratic behaviour for these intervals, reflecting the strong influence of temporal variability on this sampling strategy. For the high-intensity event, the MBE and slope strongly decrease for 1-minute intervals and longer, while the RMSE strongly increases. At these timescales the signal changes significantly, due to the decrease in rainfall intensity being relatively large compared to the noise. The peak attenuation in this case is roughly 21 dB, while the standard deviation of the noise ranges roughly between 0.15 and 0.20 dB. This illustrates the sensitivity of instantaneous sampling to signal fluctuations as a consequence of rainfall variability.

The RMSE for the theoretical events with and without noise barely increases for longer time intervals, except for instantaneous sampling with the high-intensity event. Similarly, $r^2$ is close to 1 and independent of interval length for the theoretical events. The same also holds for MBE and slope. Sampling a mean value or min and max values per time interval also seems to have a minor influence on the theoretical events, since the noiseless events, show no difference between the mean and min-max sampling strategy. Generally, this shows that each of the sampling strategies is capable of producing correct rainfall estimates, especially for the shortest time intervals. For longer time intervals, in particular the instantaneous sampling strategy does not perform as well as the other two sampling strategies. Additionally, the influence of noise on rainfall intensities cannot be neglected when using CMLs to measure rainfall.

The instantaneous sampling strategy for the 26 GHz link with noise added performs significantly worse than the other strategies and links in terms of RMSE and the $r^2$ for the low-intensity event. This is caused by the relatively low signal-to-noise ratio for the 26 GHz link in comparison to the 38 GHz link. Following from the R-k relationship, the attenuation of a 26 GHz device is lower than the attenuation of a 38 GHz device for the same rainfall intensities (e.g., for our setup a 2 mm h⁻¹ rainfall intensity results in an attenuation of 1.2 dB for a 38 GHz device and an attenuation of 0.5 dB for a 26 GHz device). Especially for low rainfall intensities, this causes that the added noise occasionally compensates for the rainfall attenuation, so that some of the attenuations are negative. In the rainfall retrieval algorithm these negative attenuations are corrected to 0 dB, also affecting the overall statistics.

It should be noted that differences in baseline power levels between sampling strategies and wet-antenna attenuation are not included in these theoretical events, while in Section 3.1 these clearly affected the rainfall intensity estimates. For example, wet-antenna attenuation makes the previously described behaviour of the instantaneous sampling strategy for the RAL 26 GHz link less likely to occur in the actual data, because the lowest attenuation levels will increase as a consequence of wet antennas.

Differences in baseline power levels are only slightly reflected in the theoretical events as caused by the added noise, which might slightly affect the median signal intensity for the computation of the baseline power levels.

Moreover, a comparison between the rainfall estimates for the noiseless signals starting at the fixed full hour (Fig. 7) and the shifted starting times (Fig. 8), shows that mean sampling strategy is most robust against different starting times of rainfall events. The instantaneous sampling strategy is most sensitive to different starting times of the events, as the spread of all the statistical metrics is large, especially for the longer time intervals and the high-intensity event. For the min-max sampling strategy, the shifting of starting times also introduces additional uncertainties in the rainfall estimates, but still outperforms the instantaneous sampling strategy.

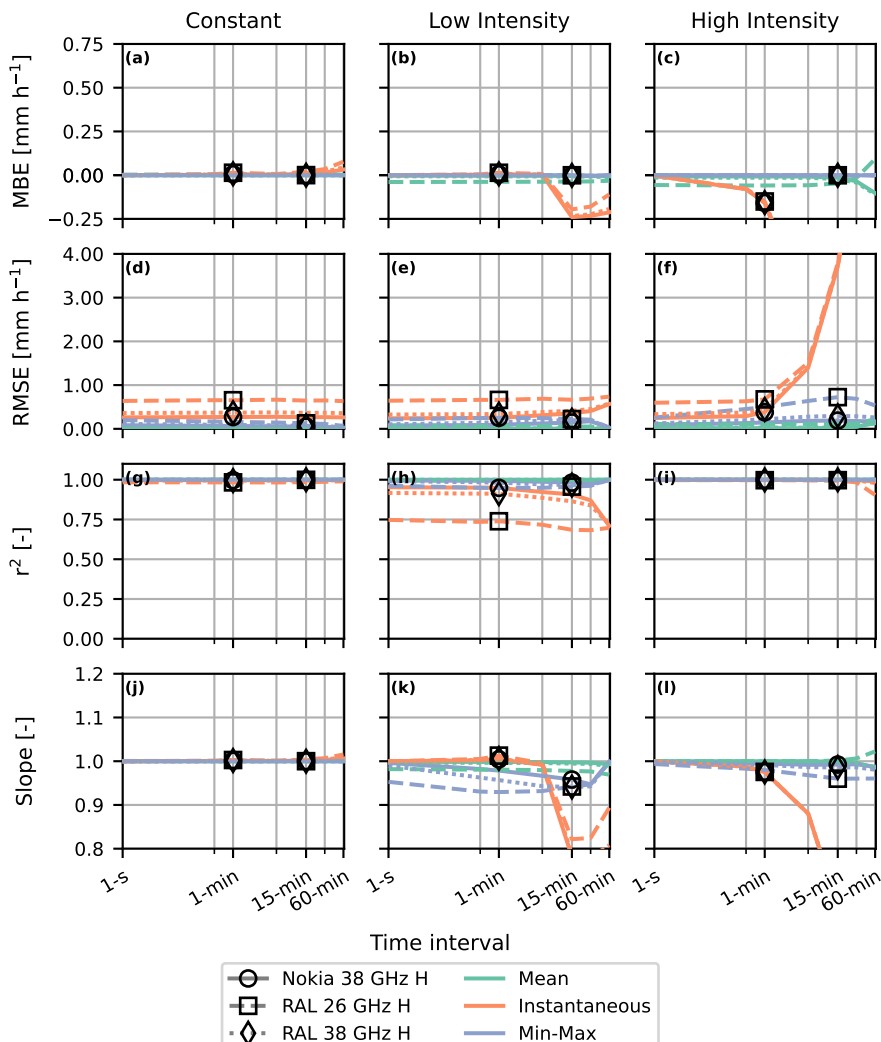

**Figure 6.** Statistical metrics as function of time interval for three theoretical rain events (Fig. 3) for all sampling methods (line colour) and added noise levels that resemble the three devices (line style). The markers indicate the values for the 15-minute min-max and 1-minute instantaneous sampling strategies.

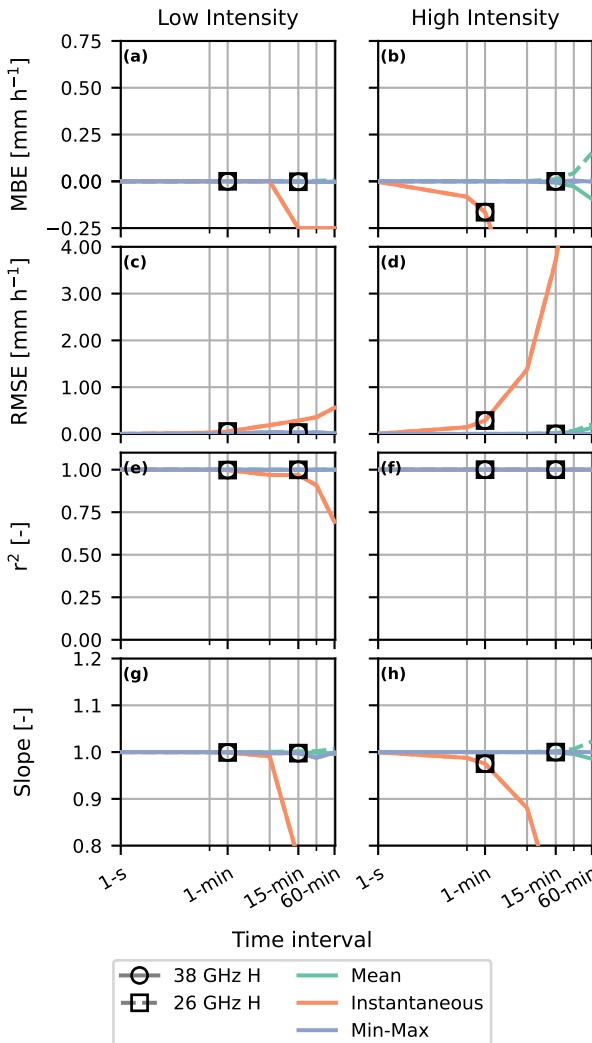

**Figure 7.** Statistical metrics as function of time interval for for a theoretical low intensity and high intensity rain event (Fig. 3) for all sampling methods (line colour) for 38 GHz and 26 GHz horizontally polarized noiseless devices (line style). The markers indicate the values for the 15-minute min-max and 1-minute instantaneous sampling strategies.

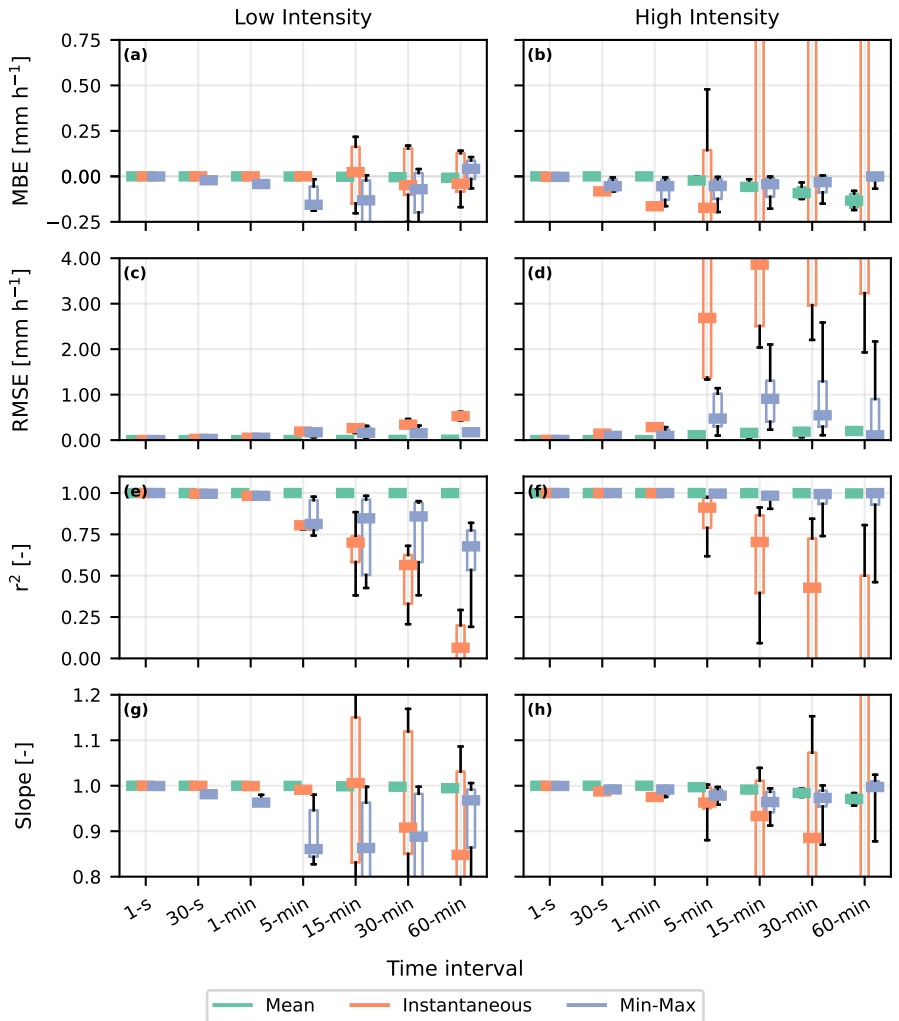

**Figure 8.** Statistical metrics as function of time interval for theoretical low intensity and high intensity rain events (Fig. 3) with a sliding 2-min start time window for all sampling methods (colour) for a 38 GHz horizontally polarized noiseless device. In the boxplot, the coloured bar is the median value, the box edges represent the first and third quartiles, and the whiskers are the 10th and 90th percentiles. The statistical metrics for a 26 GHz noiseless device reveal the same overall patterns, though with a reduced performance and increased spread for most of the metrics for the high-intensity event (not shown).

### 3.3 Influence of temporal sampling on rainfall estimates for all events

In this section, we examine how the rainfall estimates are affected by all sampling methods considered for all combinations of methods and time intervals. Figure 9 shows an example scatter plot and corresponding statistics for a time interval of 15 335 minutes. We present the statistical metrics (MBE, RMSE, $r^2$ and the slope of the linear regression line through the origin) for all considered time intervals in Figure 10. The corresponding scatter plots (Figs. S1 to S4) are provided in the Supplementary

materials. In these scatter plots, we compare the rainfall estimates for a specific device, method and time interval with the 20 Hz rainfall estimates obtained using the same device.

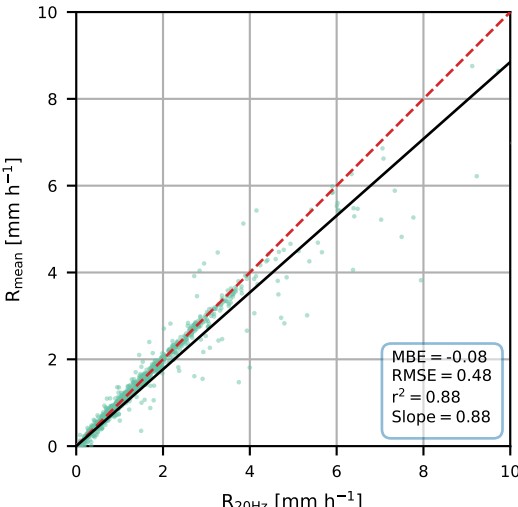

**Figure 9.** Comparison of rainfall intensities derived with mean sampling and 15 min time interval (y-axis) versus time-averaged rainfall intensities computed with the 20 Hz data (x-axis) for the Nokia Flexihopper microwave link. The red dashed line is the 1:1 line and the black line represents the best linear fit through the origin of which the slope is reported in the statistics box.

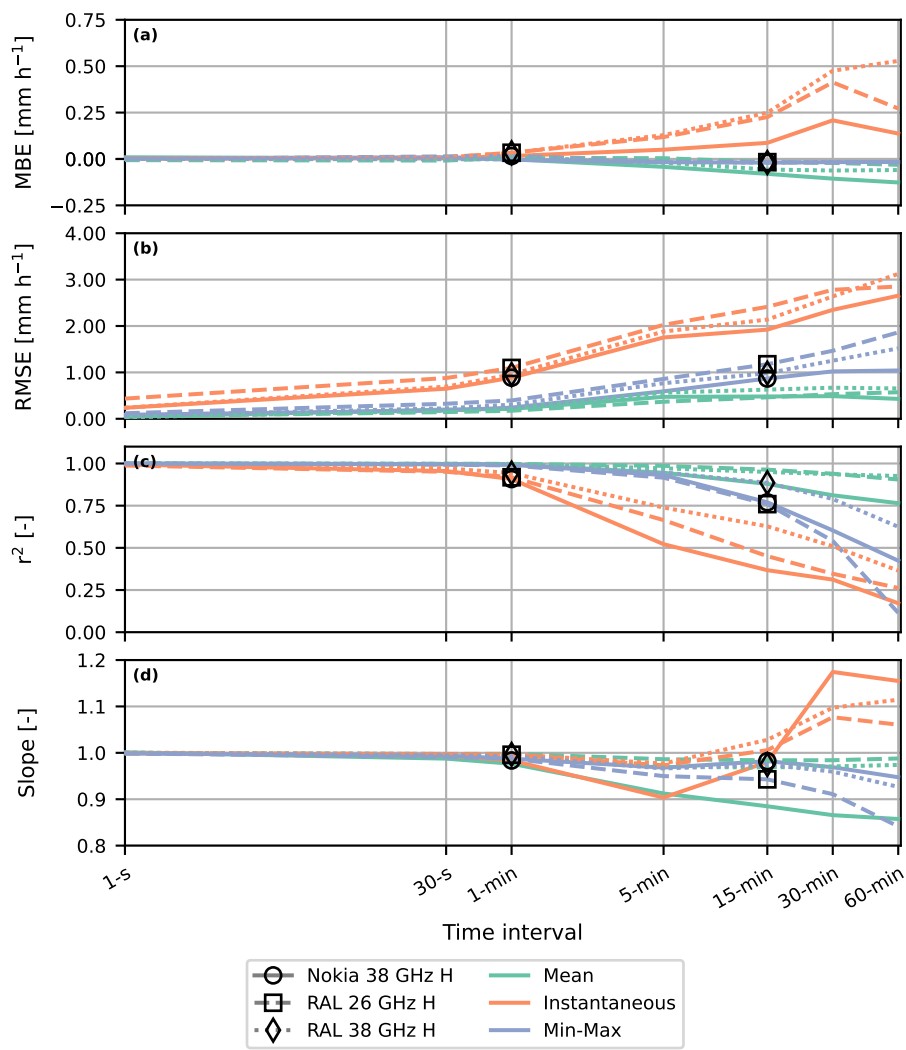

**Figure 10.** Statistical metrics as function of time interval for all devices (line style) and sampling methods (line colour). The markers indicate the values for the 15-minute min-max and 1-minute instantaneous sampling strategies. The values for the metrics are obtained from Figs. S1 to S4.

When examining the statistical metrics for all sampling methods, it is clear that the mean sampling strategy generally out-performs the other sampling methods in terms of RMSE and $r^2$. The instantaneous sampling strategy is generally outperformed by the min-max sampling strategy. Regarding MBE, the min-max sampling outperforms mean sampling, though it should be noted that part of the optimization of $\alpha$ was based on the MBE (next to the RMSE) for the entire dataset (Sect. 2). This also causes all the min-max lines in Figure 10a to be close to zero. The non-optimized min-max sampling strategy performs slightly worse than the optimized strategy, but no major differences between both occur, except for the longest time intervals (see supplementary materials).

For the shortest time intervals, absolute differences in RMSE and r$^2$ between mean and min-max sampling are insignificant (smaller than 0.1 mm h$^{-1}$ in terms of RMSE), but an increase in time interval length causes the performance of mean sampling to decrease less than for the min-max sampling, resulting in an absolute difference between the mean and min-max sampling in the order of 0.5 - 1.5 mm h$^{-1}$ for the RMSE for the Nokia and RAL links.

### 3.3.1 Mean sampling strategy

The mean sampling shows a decrease in the slope of the regression line, as the length of the time interval increases, especially for the Nokia link, culminating in a value of 0.86 for a 60-minute time interval, while we would have expected it to stay equal to 1, as roughly occurs for the RAL links. We attribute this to two opposing causes for error at long time intervals with mean sampling. The first is the reduced impact of the non-linearity of the power law, which decreases as a consequence of longer averaging time intervals resulting in lower average attenuation values (step 7 in Sect. 2.2). The second is the wet antenna attenuation, which reduces significantly on hourly timescales due to the drying of the antenna cover. In the 20 Hz data, any erroneously estimated rainfall due to wet antennas has been filtered out (i.e., set to 0 mm m$^{-1}$) based on the path-weighted disdrometer data indicating dry weather, while for long time intervals the wet-antenna attenuation is included in the computation of the rainfall intensity for the mean sampling, because of the antennas still being wet during dry weather, causing an overestimation of the rainfall intensities. For the Nokia link, the influence of wet-antenna attenuation is reduced, which results in an increased underestimation for longer time intervals, in contrast to the RAL 38 GHz link. The RAL 26 GHz link exhibits less wet-antenna attenuation than the RAL 38 GHz link, but somewhat more than the Nokia link. Moreover, at this frequency the exponent of the $R - k$ relation is below 1, so that this would cause an overestimation instead of the underestimation for the 38 GHz links (step 7 in Sect. 2.2). Still, also the mean sampling at this frequency results in an overall slope of the linear regression line through the origin below 1, though minor, which suggests that the mean sampling has a slight tendency to underestimate the rainfall intensity. Overall, this shows the potential influence that even mean sampling can have on rainfall estimates, although it is relatively small in comparison to the other sampling methods. We attribute a large part of these differences to wet-antenna attenuation and differences in estimated baseline power levels, as the behaviour of these statistics is not reflected in the theoretical events (Fig. 6 versus 10).

### 3.3.2 Instantaneous sampling strategy

For instantaneous sampling with short time intervals, the RMSE is larger than for the mean sampling method, being 0.69 mm h$^{-1}$ versus 0.17 mm h$^{-1}$, respectively, for the RAL 38 GHz link at a 30 s time interval. Yet, this spread results in a relatively low MBE of 0.01 mm h$^{-1}$ and relatively high r$^2$ of 0.99, suggesting a symmetric distribution of the residuals around the reference (Fig. 10). For both RAL links, this distribution appears to be even present at the longest time interval (Fig. 11b). We would have expected that the rainfall intensity measured at the end of a long interval does not provide any information on the other rainfall intensities during that interval, given the 30 min average duration of rainfall events in our data. The Nokia link behaves more as expected (Fig. 11a), as the spread is larger, also resulting in a lower r$^2$, i.e., 0.17 versus 0.37, at a 60-min time interval for instantaneous sampling.

Based on the individual events, it seems that the majority of the differences in performance between the devices for the instantaneous sampling strategies are related to the variations in rainfall intensity, i.e., 30 min average duration (Fig. 2), in combination with wet-antenna attenuation. This is supported by the absence of any similar differences in the theoretical events. For the shortest time intervals, the rainfall intensities, and thus measured attenuations, do not vary much in these few seconds, which results in relatively good scores. For the longer time intervals, the RAL devices are still wet, as these were found to have an average drying time of the order of 20 minutes (van Leth et al., 2018a), resulting in higher estimated rainfall intensities for both the averaged 20 Hz measurements and the instantaneous sampling. The Nokia antenna covers are already mostly dry after a few minutes (van Leth et al., 2018a), such that the estimated rainfall intensity gives very little information on the preceding rainfall intensities, resulting in the lowest MBE for the Nokia link. We suspect that this role of wet-antenna attenuation also causes the MBE to be positive for all devices compared to the other sampling strategies.

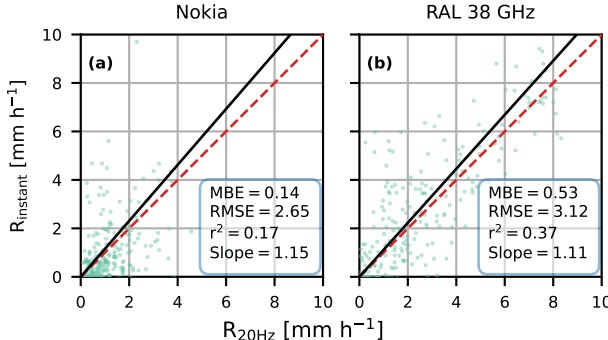

**Figure 11.** Comparison of rainfall intensities derived for the instantaneous 60-min time intervals for the Nokia Flexihopper and the RAL 38 GHz microwave links versus time-averaged rainfall intensities computed with the 20 Hz data of these devices. The red dashed line is the 1:1 line and the black line represents the linear regression line through the origin of which the slope is reported in the statistics box.

### 3.3.3 Min-max sampling strategy

For the min-max sampling with an optimized $\alpha$, the role of the minimum attenuation increases with time interval, i.e., $\alpha$ decreases from just below 0.5 to around 0.2 (Fig. 12). For the shortest time intervals, this points to a roughly symmetrical distribution of the minimum and maximum attenuations around the mean. For the longer time intervals, this suggests a positively skewed distribution of the attenuations with relatively many outliers to the maximum attenuation. For the non-optimized min-max sampling, where with increasing time intervals the maximum attenuation becomes increasingly dominant, this results in an increasing slope of the fit (Figs. S1 to S4).

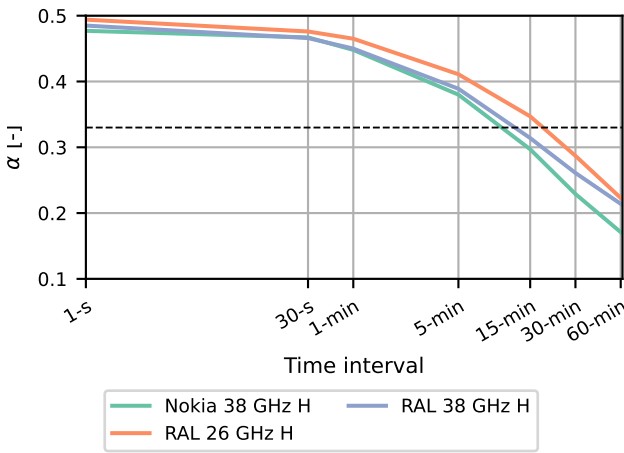

**Figure 12.** Optimized fractional contribution of the maximum and minimum attenuation, $\alpha$, as function of time interval for each device for the optimized min-max sampling method. The dashed black line represents the default RAINLINK value, being 0.33. The data is based on Figures S1 to S4.

For the 15-minute min-max sampling strategy, the optimized values of $\alpha$ for the links range between 0.30 and 0.35. This is close to the default RAINLINK value (Overeem et al., 2016b), which uses 0.33 for the contribution of minimum attenuation and maximum attenuation. This is also reflected in the RMSE, MBE, $r^2$ and slope of the fit (Figs. S1 - S4). This shows that for microwave links in similar frequency domains and precipitation climatologies, this default value is a good estimate when using 15-minute minimum and maximum signal intensities.

Moreover, for low rainfall intensities, especially below 2 mm h$^{-1}$, the rainfall retrieval algorithm for the min-max sampling strategy overestimates rainfall intensities (Fig. 13). Reasons for this seem to be twofold. Firstly, the minimum attenuation is set to 0 dB km$^{-1}$, due to the maximum received power (resulting in minimum attenuation) being higher than the baseline power level. Part of this is caused by the assumption that the median of the average of the minimum and maximum received power levels represents the baseline is not always valid, due to a skewness towards minimum power levels. For example, between 9:00 and 11:00 in Fig. 4t for the hourly time intervals, the maximum attenuations are constant, while the minimum attenuation changes between the intervals 9:00-10:00 and 10:00-11:00. Still this results in a constant rainfall intensity (Fig. 4x). Secondly, the minimum received power level (resulting in the maximum attenuation) is nearly always significantly lower than the baseline power level. This means that for low rainfall intensities, especially around and below 1 mm h$^{-1}$, the minimum attenuation is corrected (i.e., set to 0 dB km$^{-1}$), while the maximum attenuation is treated as is, preventing the min-max retrieval algorithm to work the way it has been designed. In the rainfall intensity computation, both attenuations are treated in a similar manner, giving too much weight to the maximum attenuation, causing an overestimation of the rainfall intensity. For slightly higher precipitation intensities (1-2 mm h$^{-1}$), the maximum power levels are still close to the baseline, so that barely any increase in rainfall intensity in the min-max sampling is computed, while the 20 Hz rainfall intensity estimates do increase, causing the

415 bend seen in Fig 13. For even higher rainfall intensities (>2 mm h⁻¹), both the minimum and maximum received power levels are lower than the baseline, so that the min-max sampling method can be used the way it was designed.

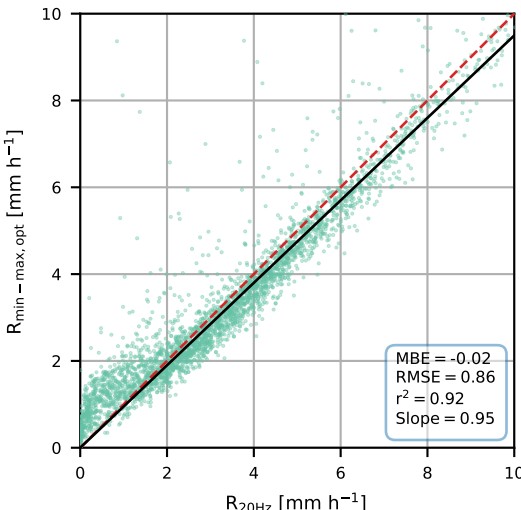

**Figure 13.** Comparison of rainfall intensities derived for min-max sampling at a 5-minute time interval (using an optimized weight) versus time-averaged rainfall intensities computed with the 20 Hz data for the RAL 26 GHz microwave link. The red dashed line is the 1:1 line and the black line represents the linear regression line through the origin of which the slope is reported in the statistics box.

### 3.3.4 Influence of signal frequency

When focusing on the influence of the signal frequency on the rainfall estimates, the RAL 26 GHz link shows a larger RMSE and lower $r^2$ for the instantaneous and min-max sampling in comparison to the RAL 38 GHz link (Fig. 10). For both sampled

variables, the RMSE is roughly constantly 0.1 to 0.3 mm h⁻¹ higher, while the $r^2$ only differs at the longest time intervals around 0.1 to 0.2. These differences can predominantly be attributed to the difference in frequency, which causes different exponents in the $R - k$ relation and minor differences in wet-antenna attenuation, as the devices do not contain any other significant differences. However, these differences in behaviour are of a smaller magnitude than the differences between the Nokia and the RAL 38 GHz links. Overall, this indicates that a reduced duration of wet-antenna attenuation and hardware reducing the

signal fluctuations can significantly reduce the influence of the selected temporal sampling strategy.

### 3.3.5 Comparison with theoretical events

Also, a general comparison between the theoretical events and all rain events in the dataset reveals the influence of wet-antenna attenuation on the performance of the rainfall retrieval algorithm. The increase of RMSE with longer time intervals found for all rain events is not reflected in the theoretical events (only minor for the min-max sampling). As discussed, we expect this

difference to be caused by the influence of wet-antenna attenuation on the RMSE. Moreover, the wet-antenna attenuation seems to have a major influence on the estimated rainfall intensities for the individual events in Sect. 3.1. Additionally, differences in

baseline power levels between sampling strategies could play a role here. These differences in the baseline in the dataset vary due to the differences in wet and dry periods between sampled intervals and strategies, and also due to longer term changes in power levels (e.g., as a consequence of temperature). This shows that the wet-antenna attenuation and differences in baseline power levels can play a relatively important role in the performance of the sampling strategy, next to the general sensitivity to sampling interval and method. This will be more elaborately discussed and put into context with previous studies in Sect. 4.

## 4   Discussion

This study aimed to determine the influence of temporal sampling strategies on measuring rainfall using microwave links. To do so, we resampled a 20 Hz signal to various temporal sampling strategies, partly mimicking sampling strategies employed by mobile network operators, and compared the resulting rainfall intensities with averaged 20 Hz rainfall intensities. This way, we were able to exclude the instrumental bias and uncertainty from the total bias and uncertainty, and focus on comparing the influence of the temporal sampling strategies.

### 4.1   Influence of sampling strategies on rainfall estimates

Our results allow to estimate the source of error for estimated rainfall intensities originating from the applied sampling strategy and puts rainfall estimates of previous and future studies in perspective in relation to their reference data. When examining the influence of time interval on estimating rainfall intensity, as expected, an increase in sampled time interval reduces the performance of the resampled rainfall estimates compared to the 20 Hz estimates. For the shortest intervals, the performance, 1 s to 1 min, often does not vary largely with respect to the reference, especially when compared to the longest intervals. An increase in resolution does not necessarily add much more information on the rainfall intensity for these short intervals, due the generally limited variability in rainfall intensity for these intervals. As described by de Vos et al. (2018), the performance of the rainfall retrieval algorithm is strongly dependent on the employed temporal sampling strategy. Moreover, Leijnse et al. (2008) stress the importance of sufficient sampling, which allows to cover the temporal behaviour of a rain event. For example, considering that a precipitation event in our dataset takes on average 30 minutes, the commonly used 15-minute sampling interval would undersample an average precipitation event, which is also reflected in our analyses, showing that the performance for almost all sampling methods decreases significantly from around 1 to 5 minutes and onwards.

As expected, a comparison between the considered sampling methods demonstrates that mean sampling generally outperforms the other methods. For the shortest time intervals, min-max sampling, especially the optimized version, performs roughly equal to mean sampling. For longer time intervals, the performance decreases, as the minimum and maximum attenuation are no longer representative for the average precipitation during the interval. When comparing the different starting times of the theoretical events, the mean sampling strategy is also most robust to potentially unfavourable starting times, while the min-max sampling strategy is more sensitive, especially for longer time intervals. Generally, the non-optimized min-max sampling strategy also performs relatively well, but a bit less than the optimized version. This decay in performance is also partly reflected in the statistical metrics of the high-intensity theoretical event, but not for the constant and low intensity event, partly caused

by the still fairly constant intensity of the low intensity event. Additionally, the min-max sampling contains a change in slope
at low rainfall intensities at longer time intervals, which we expect to be caused by the baseline level selection (i.e., the median
of the average of the minimum and maximum received power levels during the dry hours preceding a rain event). This method
for baseline determination causes the minimum attenuation (i.e., maximum power levels) during the interval to be higher than
the baseline at low rainfall intensities and be set to zero, which interferes with the algorithm as it was designed. This causes an
offset when using min-max sampling for low rainfall intensities.

Moreover, the minimum and maximum sampling strategy is mostly prone to overestimations of the rainfall intensity, especially for longer time intervals. For both the optimized and non-optimized sampling strategy, the residuals around the 1:1
line are asymmetrically distributed, especially for longer time intervals. In general, underestimations of rainfall intensity are
frequently present but relatively minor, while overestimations happen less frequently but often have a relatively large magnitude. We hypothesize that this is either caused by short high-intensity rainfall intensities during a time interval or a short and
severe reduction in signal intensity due to other causes (e.g., refraction, obstacles), which causes a relatively large maximum
attenuation, so that the estimated rainfall intensities during the entire time interval overestimate the 20 Hz rainfall estimates.
The timing of the peak intensities is often estimated well when using this sampling strategy, especially in comparison to the
instantaneous sampling strategy.

The instantaneous sampling strategy performed well for the shortest time intervals. For longer time intervals, especially 30
and 60 minutes, and high-intensity events, this strategy shows the largest sensitivity to different starting times of the theoretical
events. Also for these longer time intervals, the performance is mostly related to the amount and duration of wet-antenna
attenuation combined with the average duration of a precipitation event in our dataset. For a device with a relatively short
duration of the wet-antenna attenuation, the Nokia link, the $r^2$ was significantly less than for a device with relatively long-lasting wet-antenna attenuation, the RAL 38 GHz link. However, it is important to note that the high $r^2$ in that case is not caused
by the measurement of rainfall, but solely due to the fact that for long time intervals wet-antenna attenuation is interpreted as
rainfall in the algorithm. Potentially, an improvement in performance of the instantaneous sampling strategy could be found in
adapting the timing of the time intervals, so that the values of the received power levels represent the middle of the intervals.
Intuitively, especially for relatively large changes in rainfall intensity during a time interval, it seems that these values could be
more representative for rainfall intensities than the last value of the interval. Additionally, for high peak rainfall intensities, the
performance of the instantaneous sampling strategy, both in timing and maximum rainfall intensity, seems to be dependent on
how well single instantaneous measurements represent the entire time interval.

When comparing the conventional 15-minute min-max and 1-minute instantaneous sampling strategies, the overall statistics
are similar. For the theoretical low-intensity event with sliding window for a 38 GHz noiseless link (Fig. 8), the 15-minute
min-max sampling strategy shows a reduced performance in comparison to the 1-minute instantaneous sampling strategy. For
the high-intensity event, the performances of the sampling strategies are on average more comparable. Overall, this could imply
that in regions where low rainfall intensities are more prevalent than high intensities, it could be beneficial to use a 1-minute
instantaneous sampling strategy instead of the 15-minute sampling strategy. For the actual data (Fig. 10), the $r^2$ and the slope
for the 15-minute min-max sampling strategy are on average for all devices slightly lower (i.e., further from one) than for the

1-minute instantaneous sampling strategy. However, these differences are not of the same order of magnitude as differences with other time intervals. A reduction in sampling time interval for both sampling strategies (namely instantaneous versus min-max) would result in a larger increase in performance than the difference in performance between these sampling strategies, especially for the 15-minute min-max sampling strategy.

Previous studies demonstrated the performance of various sampling strategies for measuring rainfall with CML networks. Our results are not fully in line with Pudashine et al. (2021), who show that the min-max sampling strategy slightly outperforms the mean sampling strategy, though it should be noted that their study uses gauge-adjusted radar data as reference, which makes an objective comparison difficult. At a 15-minute sampling time interval, de Vos et al. (2019) demonstrate that min-max sampling generally outperforms instantaneous sampling in the Netherlands. Leijnse et al. (2008) compared 18 Hz sampling, averaged 15-minute sampling and instantaneous 15-minute sampling methods with each other (together with simulated microwave link data based on 15-second radar data) and showed the limitations of the instantaneous sampling compared to the other two methods. These findings are in line with our results and confirm that the differences are not solely caused by instrumental biases between the microwave links and the used reference instrument (e.g., rain gauge or radar).

## 4.2 Additional sources of uncertainties and errors

Our study shows the significant influence of wet-antenna attenuation on the performance of sampling strategies, especially with increasing sampling intervals. The rainfall intensities measured with the Nokia link, which is affected only during a relatively short period by wet-antenna attenuation, were less affected by the temporal sampling strategy than the RAL 38 GHz link, a device transmitting in the same frequency domain but with an increased wet-antenna attenuation effect. An additional difference between these devices is the reduced signal fluctuation in the Nokia link, likely caused by the different hardware employed in the Nokia link. However, these differences do not have an influence of the same order of magnitude on the raw signal. Where the hardware causes the fluctuations to reduce roughly by 0.5 dB, the additional wet-antenna attenuation for the RAL link is roughly 2 dB higher. Therefore, we attribute the largest differences between the Nokia and RAL 38 GHz links to the difference in wet-antenna attenuation.

This is confirmed when studying the statistical metrics of the theoretical rain events and comparing these with the actual rain events. The simulated Nokia and RAL links do not show significant differences for the theoretical rain events. The implemented differences in noise between the theoretical links do not result in differences between the statistical metrics similar to the actual events for the two devices in any of the theoretical rainfall events. A comparison between theoretical events with and without noise shows that the addition of noise has a significant contribution to the rainfall estimates, especially for the instantaneous sampling strategy at longer time intervals. The most significant differences between the theoretical events and the actual events can be found in wet-antenna attenuation and differences in baseline power levels between sampling strategies, which shows the importance of these two mechanisms on rainfall estimates. Additionally, it should be noted that the duration of the theoretical events is not equal to the average duration of a rainfall event in the Wageningen data (2 h and roughly 30 min, respectively), but was chosen to resemble the two individual events studied in Sect. 3.1. However, we do not expect this mismatch in timescales to have a major effect on the differences between the theoretical and actual data, i.e., the influence of wet-antenna attenuation

and different baseline power levels. Overall, this shows that knowing the attenuation caused by a wet antenna and the total drying duration would be beneficial when estimating rainfall using microwave links. Additionally, it would create more need to adjust for the instrumental bias of microwave links, for example as a consequence of different antenna covers or temperature dependence (e.g., van Leth et al., 2018a).

Different algorithms to correct for wet-antenna attenuation exist. Kharadly and Ross (2001) developed a model allowing to correct for wet-antenna attenuation, which was extended by Minda and Nakamura (2005) to be able to cope with the drying of antennas. Schleiss et al. (2013) first experimentally studied the role of wet-antenna attenuation on a 38 GHz microwave link signal during and after a precipitation event and showed that the attenuation increases exponentially during the first part of a rain event towards a maximum of 2.3 dB and also decreases exponentially afterwards. Subsequently, they proposed a model applicable without the need for additional measurements. In parallel, Overeem et al. (2013) also found a value of 2.3 dB for the wet-antenna attenuation as an average for a complete telecom network in the Netherlands. Subsequently, Overeem et al. (2016a) applied this value as a constant for correcting attenuation due to wet antennas for each microwave link. Leijnse et al. (2008) propose to use a more physics-based model to compute the wet-antenna attenuation, which uses signal frequency, antenna cover properties and rainfall intensities and seems especially useful for shorter time intervals. Graf et al. (2020) found that for a telecom network in Germany a correction based on rainfall intensity (based on Leijnse et al., 2008) outperformed a method based on the time (based on Schleiss et al., 2013) during and after a precipitation event. Similarly, Pastorek et al. (2022) compared multiple wet-antenna attenuation corrections and also concluded that corrections based on rainfall intensity outperformed other methods. Additionally, they found that these corrections can be applied to intensities obtained from sub-links with various frequencies and path lengths, and thus can also be applicable to other networks with similar antenna characteristics. Note that algorithms are applied irrespective of the number of wet antennas (0, 1 or 2). It may even be raining along the CML path, whereas antennas remain dry.

Using 20 Hz data as reference data to compare the estimated rainfall intensities has as advantage that the direct instrumental bias is excluded in our analysis. These instrumental biases are included in the comparisons done by van Leth et al. (2018a) and similar to our comparison between the 20 Hz estimates and the disdrometer (Sect. 2). Zinevich et al. (2010) describe various errors and uncertainties that arise when comparing rainfall estimates from commercial microwave links with rain gauges, for example uncertainty in the drop size distribution, wet antennas or baseline variations. Also, Leijnse et al. (2008) describe similar influences of these uncertainties and stress the need to include all potential error sources in an analysis. Our results and those from van Leth et al. (2018a) suggest that these uncertainties, especially wet-antenna attenuation, baseline variation and the non-linear effect of the power law, affect the performance of the rainfall estimates, even when using the same microwave link as reference.

Additionally, we illustrate that the magnitude of these additional biases could depend on the selected sampling method. These biases can be separated into a directly affected part and an indirect part. Rainfall intensities are most often directly affected when choosing an instantaneous sampling method, especially for longer time intervals. Moreover, wet-antenna attenuation and the role of the exponent in the $R - k$ relation also seem to play a direct role in the performance of the selected sampling strategy, especially when considering the individual rain events and compare the theoretical events with the whole dataset.

Indirectly, we suspect that observed biases, especially between sampling strategies, can occur mostly due to variations in baseline power levels between selected sampling methods. The latter group is hard to consider beforehand and will most often be an unavoidable source of uncertainty for other studies. In general, our efforts allow future studies to estimate the uncertainty of their observed rainfall intensities as a consequence of the chosen sampling strategy and potentially uncover the instrumental bias of these links.

### 4.3 Extrapolation to CML networks

When extrapolating our results to CML networks, first of all it should be noted that not all CML networks contain similar frequencies and link lengths. Next to a change in the parameters for the $R - k$ relationship, our results show a slightly increased RMSE and reduced r$^2$ for the RAL 26 GHz link compared to the RAL 38 GHz link. These devices operate similarly, except at a different frequency, which results in different $a$ and $b$ parameter values in the $R - k$ relationship. For 38 GHz, $b$ is larger than 1, while for 26 GHz, $b$ is smaller than 1. Also, between the devices a minor difference in wet-antenna attenuation can be observed, which could be a consequence of differences in frequency. Therefore, we attribute the observed differences to differences in frequency and wet-antenna attenuation (possibly as a consequence of frequency). For an increase in link length, we expect a reduced sensitivity of the sampling strategies to increasing time intervals, while also the differences between sampling strategies decrease. As suggested by Leijnse et al. (2008), this is caused by the increase in characteristic timescales of the path-averaged rainfall intensities when increasing link lengths. Similarly, Berne and Uijlenhoet (2007) showed for longer link lengths a decrease in uncertainty in rainfall estimates and, moreover, a reduction in sensitivity to sampling effects. For shorter link lengths, an opposite behaviour is expected. This could, for example, imply that part of the overestimations found for short 38 GHz links are not only a consequence of wet-antenna attenuation (Fencl et al., 2019), which does play a significant role, but also of sampling strategy.

The International Telecommunication Union (ITU-R, 2005) reports values for the $a$ and $b$ parameters, though several studies found location dependent parameter values. Leijnse et al. (2007b) showed differences up to 10% in the $b$ parameter compared to the ITU recommendations, when deriving these parameters using raindrop size distributions measured in the Netherlands by Wessels (1972). In this study, we used the $a$ and $b$ parameter values as reported by van Leth et al. (2018a), who determined these based on the five disdrometers along the path. Yet overall, Chwala and Kunstmann (2019) illustrated that the $a$ and $b$ parameters are relatively independent of the drop size distribution at low frequencies, especially below 40 GHz. Additionally, Berne and Uijlenhoet (2007), Overeem et al. (2011) and Overeem et al. (2021) demonstrate that using $a$ and $b$ parameter values which are not specifically calibrated to the area of application only results in minor errors in the rainfall estimates. This is likely a consequence of the near-linear relation between attenuation and rainfall intensity.

Other differences between our results and actual CML networks can be found in the difference in polarization. Most microwave links in CML networks are vertically polarized, instead of the horizontal polarization that we study. For this specific study, however, there are no significant differences between the vertical and horizontal polarization of the RAL 38 GHz link, probably due to the fact that our reference data, i.e., the 20 Hz estimates, have the same polarization, and thus attenuation at the same frequencies. Network operators choose for the vertical polarization, due to the oblate raindrop shape which induces

an increased attenuation for horizontal polarization compared to the vertical polarization. For other purposes than this study, it can be important to consider the polarization of the device.

Furthermore, CML networks often employ a signal power quantization (i.e., the discretization of the signal intensity) of 1 dB, while the data we use in this experiment has been designed to prevent the power quantization effect on the rainfall estimates. Leijnse et al. (2008) demonstrate that power quantization can have a significant effect on the estimated rainfall intensities when using CML networks, especially for low rainfall intensities. Ostrometzky et al. (2017) show that min-max sampling combined with the quantization effect can lead to significant biases for rainfall retrieval. Chwala and Kunstmann (2019) show that the quantization effect limits the minimal detectable rainfall intensity. Future studies could complement our study by focussing on the influence of power quantization on the rainfall estimates using the same dataset. Additional uncertainties arise from the wet-dry classification. We used disdrometer observations to classify the weather as dry or wet, while these nearby in-situ data are usually not available in a CML network, which could for example result in baseline variation uncertainties. For example, Messer and Sendik (2015) provide an overview of various wet-dry classifications and how this affects the baseline power levels.

## 5  Conclusions

In this study, we examined the influence of temporal sampling strategies on estimating rainfall using microwave links based on a dataset containing three different microwave links obtained by van Leth et al. (2018a). We compared the mean, instantaneous and min-max sampling strategies and various time intervals ranging from 1 s to 60 min with 20 Hz rainfall estimates of the same device, allowing us to exclude the direct instrumental bias in this comparison. Some of the applied sampling methods are also employed in current CML networks. Moreover, we used disdrometer data for a wet-dry classification to determine the baseline power levels.

In general, our results show that an increase in the duration of the sampled time interval reduces the performance of the resampled rainfall estimates. The instantaneous sampling method is most sensitive to this dependency on sampling interval, while the mean sampling is most robust. This is also in line with expectations, as the instantaneously sampled power levels at longer time intervals do not contain any information on the rainfall during the preceding time interval, so only describe what occurs exactly at the moment of sampling. However, for the mean sampling still differences with respect to the reference can occur due to the exponent in the $R - k$ relationship being different from one. For the shortest time intervals, both instantaneous and mean sampling perform roughly equal, except that instantaneous sampling shows a larger RMSE than mean sampling. Similarly, for the shortest time intervals, min-max sampling performs roughly the same as mean sampling, but with an increase in time interval the performance of min-max sampling shows a slightly larger decay than mean sampling (but smaller than instantaneous). For the common 15-minute sampling strategy and the former Nokia CML, mean and min-max sampling strategies clearly outperform instantaneous sampling. To a lesser extent, the mean sampling strategy outperforms the min-max sampling strategy, except for the MBE value, due to the optimization based on the MBE.

The partitioning between the minimum and maximum attenuation, represented by $\alpha$ in the rainfall retrieval algorithm, shows a decrease (i.e., more influence of the minimum attenuation) with an increase in time interval when optimizing this parameter, as for longer time intervals the maximum attenuation is less representative for that entire interval. For all min-max sampling time intervals, a change in slope occurs roughly below 2 mm h$^{-1}$, which induces a positive offset for these low intensities when employing a min-max sampling method. Generally, this sampling strategy is mostly prone to slightly underestimate or largely overestimate individual rainfall intensities, which on average results in a relatively good fit. The large overestimations are caused by the sensitivity of the maximum attenuation to incidental high attenuations, either as a result of short high-intensity rainfall episodes or due to an outlier in maximum attenuation.

When comparing devices, the Nokia Flexihopper link, which was formerly part of a CML network, transmitting at 38.2 GHz, outperforms the other devices. This device mostly differs from the other two devices, the RAL 38 and 26 GHz links, in terms of reduced magnitude and duration of wet-antenna attenuation and is designed with hardware that reduces signal fluctuations. Of these, the wet-antenna attenuation seems to dominate over the reduced noise, so that we attribute a significant part of the differences between the Nokia Flexihopper and RAL 38 GHz links to the wet-antenna attenuation. The RAL 38 GHz device is specially sensitive to wet-antenna attenuation, which makes that even for long time intervals the RAL 38 GHz link still has a relatively high r$^2$ compared to the Nokia link. If we compare the RAL 38 GHz link to the 26 GHz link, the RMSE and r$^2$ are higher and lower, respectively, for the 26 GHz link. This is mostly caused by the somewhat increased uncertainty for the RAL 26 GHz link, which we attribute to the difference in frequency. Also, the difference in exponent of the $R - k$ relationship between 26 GHz ($b = 0.95$) and 38 GHz ($b = 1.05$) could contribute to the found differences.

Additionally, a comparison of the results with theoretical events reveals the influence of wet-antenna attenuation and possibly variations in baseline power levels between sampling strategies. In these theoretical events, the minor differences in statistical metrics between the signal mimicking the Nokia link and that mimicking the RAL 38 GHz link can be attributed to noise levels. In general, even between all the sampling strategies, minor differences occur for the theoretical events, except when shifting starting times, to which the instantaneous sampling strategy shows a large sensitivity, especially for longer time intervals and high-intensity events. Other processes, of which wet-antenna attenuation being the most significant, are excluded in these theoretical events. This illustrates the significant influence wet-antenna attenuation during rain events can have on the rainfall estimates, for all sampling strategies. On individual rainfall event levels, additional differences in rainfall estimates can arise due to variations in baseline power levels between devices and sampling methods. Overall, our study illustrates the influence a selected sampling strategy and related effects can have on the estimated rainfall intensity using microwave links, but does leave the instrumental bias to consider for future studies.

*Author contributions.* LDvdV carried out the research under the supervision of MCG, RH, AO and RU. LDvdV performed the data analysis with input from all co-authors. LDvdV prepared the manuscript, with contributions from all co-authors.

*Competing interests.*   The authors declare that they have no competing interests

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
