# Peer review of "Measuring rainfall using microwave links: the influence of temporal sampling"

_EGUsphere, 2023_

## Author Comment (AC1)

**Dear Referee,**

**We would like to thank you for taking the time to review our paper and your suggestions, which definitely helped to improve the quality of the manuscript. We reply to your comments below. Our response to the comments appears in bold and revised text as** *italic*.

Minor suggestions/remarks:

- Line 45 or 57: One sentence on CML network density and changes over time and location might highlight their relevance
  **We agree that adding a sentence on CML networks highlights their relevance. We added the following sentence at L57 (line numbers refer to the original manuscript):**
  *Moreover, the number of CMLs operating worldwide in the 6-56 GHz range, which are most useful for rainfall estimation, is expected to grow from 4.6 million in 2021 to 6 million in 2027 (ABI research, 2021).*

- Line 140: Why the decision for 30 seconds? What are the effects of another threshold on your study?

  **The disdrometer data is provided with a 30-second resolution. To refrain ourselves from making any assumptions about when a rainfall event is continuous or not, we decided to use a single timestep as threshold. We added as follows:**

  *In this figure two contiguous rainy periods, which are separated by a single 30-second dry time interval, are counted as separate events and not combined into a single event. To refrain ourselves from making any assumptions about when a rainfall event is continuous or not, we decided to use a single timestep in the available disdrometer data as threshold.*

- Line 200: As RMSE and MBE are explained it might be good to also explain r2
  **We have added the r² equation in step 7, where it is mentioned for the first time.**

$$r^2 = 1 - \frac{\sum (R_{obs} - R_{20Hz})^2}{\sum (R_{obs} - \overline{R_{obs}})^2},$$

  **This comment made us realise that we made a small mistake in the calculation of r². Instead of using the 20 Hz observation to compute the r2, we used the predicted value by the linear regression model. This causes the r² values in figure 9 to be somewhat lower, but does not affect the overall conclusions. See:**

[Figure]

**The markers at 1-min instantaneous and 15-min min-max sampling strategies are added after comments of reviewer number 2.**

- Line 240: If possible more information on this filter would be beneficial. If not, it would be good to state this.
  **We don't know exactly what kind of filter this is, or perhaps the internal hardware is designed differently (see comments reviewer 2). Therefore, we rephrased all references to this filter as follows:**

  *L106-110: The employed frequencies for the Nokia and RAL 38 GHz links are close, hence exhibit similar electromagnetic characteristics, but do not interfere with each other. However, these devices were found to give a different response, likely due to the internal hardware in the Nokia link being designed differently, reducing the high-frequency fluctuations in the signal, while the RAL link has a different antenna cover than the Nokia link, which affects the distribution of water remnants on the cover (see van Leth et al., 2018a).*

  *L242-243: These high-frequency fluctuations in the signal are roughly reduced by 0.5 dB, which is likely caused by the different internal electronics in the Nokia link.*

*L375-376: Overall, this indicates that a reduced duration of wet-antenna attenuation and hardware reducing the signal fluctuations can significantly reduce the influence of the selected temporal sampling strategy.*

*L447-451: An additional difference between these devices is the reduced signal fluctuation in the Nokia link, likely caused by the different hardware employed in the Nokia link. However, these differences do not have an influence of the same order of magnitude on the raw signal. Where the hardware causes the fluctuations to reduce roughly by 0.5 dB, the additional wet-antenna attenuation for the RAL link is roughly 2 dB higher. Therefore, we attribute the largest differences between the Nokia and RAL 38 GHz links to the difference in wet-antenna attenuation.*

*L555-556: This device mostly differs from the other two devices, the RAL 38 and 26 GHz links, in terms of reduced magnitude and duration of wet-antenna attenuation and is designed with hardware that reduces signal fluctuations.*

- Line 242: Why more fluctuations?
  **When comparing Fig. 4p and x, we observe that the estimated rainfall intensities fluctuate more for RAL 26 GHz than the RAL 38 GHz for the min-max sampling strategy. This could be caused by the different exponents in the R-k relation. We rephrased as follows:**
  *Compared to the RAL 38 GHz link, the RAL 26 GHz link shows slightly more fluctuations in the estimated rainfall intensities for the min-max sampling strategies, which could possibly be caused by the different exponent in the R-k relation.*

Minor technical remarks:

- Line 112: 'as remnant' seems a bit redundant
  **We agree on that. We removed this from the text.**
  *On the RAL cover water droplets form once it gets wet, which induces a more significant attenuation of signal intensity than the water film that forms on the Nokia cover as remnant after getting wet.*

- Figs. 4 and 5. The color between 1,s, 5min and 60min has low contrast, making it difficult to see, especially for the small figures.
  **We agree that this was indeed hard to see. We adapted the figure by giving the 60-minute line a slightly darker color. See figure 5 as example:**

[Figure]

***Figure 5***. *Comparison of received (solid) and baseline (dashed) power levels (a-d, i-l, q-t) and retrieved rain rates (e-h, m-p, u-x) during a high-intensity precipitation event on 21 June 2015 obtained with the Nokia (a-h), RAL 38 GHz (i-p) and RAL 26 GHz (q-x) microwave links for all sampled variables and the 1-second, 5-minute and 60-minute time intervals. Grey areas indicate dry periods based on disdrometer data.*

- Fig. 7: In the legend move 'mean' to the right column

  **We have moved the mean to the right column.**

[Figure]

**The markers at 1-min instantaneous and 15-min min-max sampling strategies are added after comments of reviewer number 2.**

- Line 375: Also, a general

**We agree. We added the comma:**
*Also, a general comparison between the theoretical events and all rain events in the dataset reveals the influence of wet-antenna attenuation on the performance of the rainfall retrieval algorithm.*

---

## Author Comment (AC2)

**Dear Referee,**

**We would like to thank you for taking the time to review our paper and for all your constructive suggestions, which definitely helped to improve the quality of the manuscript. We reply to your comments below. Our response to the comments appears in bold and revised text as** *italic***.**

Main comments:

1. There is a multitude of sampling strategies that are investigated in this manuscript. That results in a bit too many individual results of which not all are equally important. Almost all larger CML datasets do use either 15-minute min-max sampling or instantaneous sampling at 1 minute or faster (10 seconds is also common nowadays). The later does require a dedicated data acquisition system. Thus one very interesting question is, if it is worth to convince CML network operators to upgrade their data collection system to provide 1-minute instantaneous data instead of 15-minute min-max. Hence, I suggest to add a dedicated comparison of these two (or three if including 30sec or 1sec) different sampling strategies in the results and also in the discussion section. Parts of the results section could potentially also be reduced to increase the focus of the manuscript.

    **We agree with the reviewer that it is important to show how the more common sampling strategies perform. However, the aim of this paper is to provide a systematic overview of the consequences of different sampling strategies. Therefore, we chose to not only select the most common sampling strategies, but also less widely (or not at all) used sampling strategies. This shows the sensitivity of the specific sampling strategy to increasing (or decreasing) the temporal resolution or changing the sampled variable. To emphasise this objective, we added the following to the text:**

    *L94-95 (line numbers refer to the original manuscript): Thus, using this method, we are able to estimate the actual errors and uncertainties due to the sampling strategy.* *We do so for commonly selected sampling strategies (e.g., a minimum and maximum intensity per 15 minutes) as well as less or never selected sampling strategies, which allows to systematically illustrate the sensitivity of the measured rainfall intensities to the various sampling strategies.* *Overall, this could help to estimate the effects of the strategies set by mobile network operators or help to choose an optimal strategy when estimating rainfall intensities using CMLs.*

    **To add some emphasis to the 15-minute min-max and 1 minute instantaneous sampling strategies, we added to the overview statistics figures markers for these strategies. For example:**

[Figure]

*Figure 9. Statistical metrics as function of time interval for all devices (line colour) and sampling methods (line style). The markers indicate the values for the 15-minute min-max and 1-minute instantaneous sampling strategies. The values for the metrics are obtained from Figs. S1 to S4.*

**Please note that a comment from reviewer 1 made us realise that we made a small mistake in the calculation of r². Instead of using the 20 Hz observations to compute the r², we used the predicted value by the linear regression model. This causes the r² values in figure 9 to be somewhat lower than in our original manuscript. However, this does not affect the overall conclusions.**

2.  The result section 3.3 lacks focus or at least lacks structure. I suggest to add sub-subsection or to split up section 3.3. The discussion section would also be much easier to comprehend with some subsections and maybe also increasing the focus of the content.
    **We thank the reviewer for this suggestion. The order of the Results section was based on sampling strategy, but this may not have been entirely clear. Therefore, we split the subsection into multiple sub-subsections to guide the reader through the subsection. These sub-subsections are based on the sampling strategy and other observations (i.e., the titles**

**are** *Mean sampling strategy, Instantaneous sampling strategy, Min-max sampling strategy, Influence of signal frequency, and Comparison with theoretical events).*

**Additionally, we placed the figures closer to the reference in the text. We realize that in a publishing process, this will be reshuffled again, but we will pay attention during the formatting process to make sure that figures are close to their in-text reference.**

**In the discussion, we also added subsections:** *Influence of sampling strategies on rainfall estimates, Additional sources of uncertainties and errors, and Extrapolation to CML networks*

3. If the temporal alignment of the theoretical events is always fixed with the full hour (which is not clear form the text, but it looks like that in Fig. 3), I strongly suggest to redo this analysis with random starting times of the events because this very likely has an effect, in particular for longer sampling intervals and (I guess) in particular for instantaneous sampling.

    **We agree that always starting the theoretical events at the full hour might have a significant effect on the statistics. To investigate this, we applied a shift in the start times of the theoretical events without noise, of which we can use the results to extrapolate to the events with noise, given the assumption that the added noise does not interact with the changes in start time. We created 30 events of which we shift the start time by 2 minutes per event (i.e., we get start times shifted from the original by 2,4,6…..60 minutes). In the methods we added as follows:**

    *Additionally, we study the influence of the starting times of the theoretical rain events on the statistical metrics. To do so, we create 30 theoretical events each with a shift in starting time compared to the original theoretical events. These shifts are uniformly distributed between 2 and 60 minutes (i.e., a 2-min interval). This allows us to estimate the robustness of the sampling strategies against (possibly unfavourable) starting times of rain events.*

    **In the Results, we added the following:**

    *Moreover, a comparison between the rainfall estimates for the noiseless signals starting at the fixed full hour (Fig. 7) and the shifted starting times (Fig. 8), shows that mean sampling strategy is most robust against different starting times of rainfall events. The instantaneous sampling strategy is most sensitive to different starting times of the events, as the spread of all the statistical metrics is large, especially for the longer time intervals and the high-intensity event. For the min-max sampling strategy, the shifting of starting times also introduces additional uncertainties in the rainfall estimates, but still outperforms the instantaneous sampling strategy.*

[Figure]

**Figure 8**. *Statistical metrics as function of time interval for a theoretical low intensity and high intensity rain event (Fig. 3) with a sliding 2-min start time window for all sampling methods (color) for a 38 GHz horizontally polarized noiseless device. In the boxplot, the colored bar is the median value, the box edges represent the first and third quartile, and the whiskers are the 10th and 90th percentiles. The statistical metrics for a 26 GHz noiseless device reveal the same overall patterns, though with a reduced performance and increased spread for most of the metrics for the high-intensity event (not shown).*

**In the Discussion we added:**
*As expected, a comparison between the considered sampling methods demonstrates that mean sampling generally outperforms the other methods. For the shortest time intervals, min-max sampling, especially the optimized version, performs roughly equal to mean sampling. For longer time intervals, the performance decreases, as the minimum and maximum attenuation are no longer representative for the average precipitation during the interval. When comparing the different starting times of the theoretical events, the mean sampling strategy is also most robust to potentially unfavourable starting times, while the min-max sampling strategy is more sensitive, especially for longer time intervals. Generally, the non-optimized min-max sampling strategy also performs relatively well, but a bit less than the optimized version…..(paragraph continues)*

*The instantaneous sampling strategy performed well for the shortest time intervals. For longer time intervals, especially 30 and 60 minutes, and high-intensity events, this strategy shows the largest sensitivity to different starting times of the theoretical events. Also for these longer time intervals, the performance is mostly related to the amount and duration of wet-antenna attenuation combined with the average duration of a precipitation event in our dataset. For a device with a relatively short duration of the wet-antenna…..(paragraph continues)*

**In the Conclusions we added:**
*Additionally, a comparison of the results with theoretical events reveals the influence of wet-antenna attenuation and possibly variations in baseline power levels between sampling strategies. In these theoretical events, the minor differences in statistical metrics between the signal mimicking the Nokia link and that mimicking the RAL 38 GHz link can be attributed to noise levels. In general, even between all the sampling strategies, minor differences occur for the theoretical events, except when shifting starting times, to which the instantaneous sampling strategy shows a large sensitivity, especially for longer time intervals and high-intensity events. Other processes…..(paragraph continues)*

4. There is no differentiation between different rain rates for the analysis based on real data. But different sampling strategies might have a higher impact for heavy short duration rainfall events. Hence, it would be good to also see the differences of the sampling strategy for high rain rates. That would also allow to draw conclusion about the performance in other rainfall climates, e.g in the tropics where heavy rainfall is much more common than in Central Europe. Unfortunately, the plots in the supplementary material are cut off at 10 mm/h and are also hard to interpret for the fewer and less visible high rain rates. I suggest to add an analysis for high rain rates or for events that have a high peak rain rate (based on disdrometer data). Maybe this can be accomplished by just subsetting the data for Fig 9 to have the same figure but with data only for high rain rate events. If the authors do not want to add this analysis, they should at least discuss the impact of high rain rate events on the sampling strategies in the discussion section.

**We agree that heavy short-duration rainfall events are also important to consider. However, in this paper we aim to provide an overall overview of the influence of the sampling strategies on rainfall measurements for a broad range of rainfall intensities. Therefore, we decided originally not too include such an analysis.**

**To also show the higher rainfall intensities, the maximum range of the plots in the supplementary materials has been increased from 10 mm h$^{-1}$ to 20 mm h$^{-1}$. In the main text, we have not applied these new axis limits, because the considered sampling strategies don't have (m)any estimated intensities above 10 mm h$^{-1}$.**

**For the individual events, which are a high and low-intensity event, we added a description on the influence of sampling strategy on high-intensity events:**

*Generally, the instantaneous and min-max sampling strategies seem to be more prone to errors in retrieving high rainfall intensities. For the event on 21 June, the mean sampling strategy results, as expected, in decent estimates of the evolution of the rainfall event on average, both in timing and intensities (though the peak rainfall intensities are obviously averaged out). The instantaneous sampling strategy seems to be most sensitive to an increase in the length of the time interval, because its performance depends on the representativeness of a single measurement for the whole time interval. The timing and intensity of the estimated peak rainfall intensity for short time intervals are often relatively good, due to the large signal-to-noise ratio, while for larger time intervals the sensitivity heavily increases. For the min-max*

*sampling strategy, the timing of the peak intensity is generally well-captured, but the estimated peak rainfall intensities can be inaccurate. Additionally, for this specific case, this method strongly overestimates the rainfall sum for the 60-minute interval, due to the peak taking place around the full hour, so that two subsequent intervals cover this peak.*

**Also, in the Discussion we added the following:**

*Moreover, the minimum and maximum sampling strategy is mostly prone to overestimations of the rainfall intensity, especially for longer time intervals. For both the optimized and non-optimized sampling strategy, the residuals around the 1:1 line are asymmetrically distributed, especially for longer time intervals. In general, underestimations of rainfall intensity are frequently present but relatively minor, while overestimations happen less frequently but often have a relatively large magnitude. We hypothesize that this is either caused by short high-intensity rainfall intensities during a time interval or a short and severe reduction in signal intensity due to other causes (e.g., refraction, obstacles), which causes a relatively large maximum attenuation, such that the estimated rainfall intensities during the entire time interval overestimate the 20 Hz rainfall estimates. The timing of the peak intensities is often estimated well when using this sampling strategy, especially in comparison to the instantaneous sampling strategy.*

*(at the end of the subsequent paragraph)….. Additionally, for high peak rainfall intensities, the performance of the instantaneous sampling strategy, both in timing and maximum rainfall intensity, seems to be dependent on how well single instantaneous measurements represent the entire time interval.*

Specific comments:

- L7: „…averaged values are preferred". Why is that? Is this a conclusion of this work or from previous work?
  **Based on your comment on L82, we agree that you would want the highest possible temporal resolution data, because this will give you the most information. We removed the average values part:**
  *For rainfall research purposes, often high temporal resolutions  are preferred.*

- L17: Not sure what „In this" means here. Maybe rephrase.
  **"In this" refers to the sensitivity of the mean sampling to the reduction in temporal resolution. Indeed this was not clear from the text, therefore we rephrased as follows:**
  *Even the mean sampling strategy, which generally performs best of all strategies, is sensitive to this reduction in temporal resolution and could lead to significant underestimations. This sensitivity of the mean sampling to the temporal resolution seems to be largely due to the non-linear relation between attenuation and rainfall. The min-max sampling strategy is mostly prone to minor underestimations or large overestimations of the path-averaged rainfall intensities.*

- L22: „…attenuation due to wet antenna not only…but also has a significant influence on the rainfall retrieval algorithm". It is not clear to me how the wet antenna attenuation has influence on the retrieval algorithm. In what sense? Choice of parameters? Please be a bit more precise here.

**We meant the performance of the rainfall retrieval algorithm in this case, which is mostly affected for devices with a long lasting wet-antenna attenuation at long sampling time intervals. We added that to the sentence:**

*Moreover, our results, including a comparison with theoretical events, show that the attenuation due to wet antennas not only affects the comparison between the rainfall estimates obtained with a microwave link and another reference instrument, but also has a significant influence* on the performance of the *rainfall retrieval algorithm, especially for devices with relatively long duration of the wet-antenna attenuation combined with the longer sampling time intervals.*

- L28: What is so special about this old paper Niemczynowicz (1988) that it is added here? I am not saying that it does not belong here. It is indeed impressive to look at the plots in that paper considering the computational resources available 30 years ago. Was this the first paper to do this kind of analysis? Is yes, that could also be highlighted.
  **We added multiple references to this statement to stress the importance of the statement. We selected three different references from different periods to illustrate that this topic has already been studied for a relatively long period. To clarify that these are examples of research on this topic, we changed the statement as follows:**
  *…and urban runoff estimates are highly dependent on rainfall estimates (*e.g., *Berne et al., 2004; Cristiano et al., 2017; Niemczynowicz, 1988)...*

- L38: One main drawback of satellite products is also the latency at which they become available, in particular if they are merged products.
  **Thank you for this suggestion. We added this as follows:**
  On a global scale, including the oceans, satellites provide rainfall
  information, but for hydrometeorological applications these can come at a too low spatial and temporal resolution combined with too high uncertainty and bias, partly dependent on, for example, terrain complexity, aridity and season (Maggioni et al., 2018; Rios Gaona et al., 2017). *Additionally, satellite rainfall products, especially merged products, have a relatively long latency (e.g., IMERG has about a 4 hour latency for the earliest run; NASA, 2024).*

- L48: „when considering aggregation scales that are too large for point measurements" I find this formulation a bit confusing. Do you mean large spatial or temporal aggregation? And how is that related to points measurements? Please rephrase.
  **This is indeed not clearly formulated. We rephrased the sentence as follows:**
  *…, CMLs are especially useful when considering* spatial *aggregation scales that are too large to cover entirely with point measurements. Also, for many applications spatial rainfall estimates on scales in the order of a couple kilometres are more relevant than point measurements.*

- Equation 1: It seems to be more common to write k = aR^b. Hence, the authors might want to change that. But this is of course a matter of definition.
  **We chose to use the same definition as van Leth et al., (2018), since our research follows on their study and also makes use of their parameter values in this relationship. Additionally we estimate R based on k, so therefore we leave this as is.**

- L69: Papua New Guinea is not in Africa!
  **We made a mistake and indeed mean Papua New Guinea. We changed this as follows:**

*Since then, studies have been performed in Europe (Czech Republic, France, Germany, Italy, Luxembourg, Sweden and Switzerland), Africa (Burkina Faso, Kenya and Nigeria), South America (Brazil), Asia (Lebanon, Pakistan and Sri Lanka) and Oceania (Australia and Papua New Guinea) (see Chwala and Kunstmann, 2019, for a partial overview).*

- L82: Why is it not preferred to keep the instantaneous data for high temporal resolution sampling for research purposes? One can always average the data later.
  **We agree that ideally, you would want the highest possible temporal resolution data, because this will give you the most information. We removed the average values part:**
  *For research purposes, data with a high temporal resolution in combination with averaged values is often preferred, so that the temporal sampling resolution is higher than the dominant timescales of rainfall.*

- L90: Was one CML part of an operational network during the data collection? (Update: It is clear from text later on that it was formerly an operational CML. Maybe rephrase here.)
  **This is indeed not clear from the sentence. We changed the sentence as follows:**
  *One of these links has formerly been employed in an operational CML network.*

- L109: What is the cut-off frequency of the filter of the Nokia CML? How does that affect the 20 Hz sampling? A related question is also the effect of the bandwidth of the two systems. Higher bandwidth results in higher noise in the receiver. Can this also be an effect here?
  **See our reply to your comments on L445.**

- Figure 2 and related text: What is the purpose of the analysis of event duration? It is not clear from this section. If this analysis is just there to give an idea of the rainfall distribution it might also be interesting to show some other quantities (rainfall sum, q99 rain rate, …) in additional subplots (maybe just one in addition) of Fig2.
  **We show this data to give an idea of the rainfall distribution. We added a boxplot for rainfall intensity. See:**

[Figure]

*Figure 2. Boxplots describing the duration of rainfall events per month in 2015 (a) and all rainfall intensities per month in 2015 (b). In the boxplots, orange is the median and green is the mean duration or intensity. The box edges represent the first and third quartile, and the whiskers are the 10th and 90th percentile. The values are computed using disdrometers installed along the 2.2 km microwave link path. In the duration statistics, events shorter than*

*5 minutes are excluded. For the rainfall intensities, all rainfall intensities larger than 0.1 mm h⁻¹ are used.*

**We also added in the text:**
*The rainfall intensities show in general higher peak intensities during summer than winter. Due to the power outage between 7 and 25 August, the data for that month can be less reliable.*

- L217: „The latter two events resemble two individual events…". I do not get why these two are individual events. Please explain and maybe rephrase in the manuscript. (Update: After reading section 2.3 I know understand. I suggest to write „…resemble the two real rain events that are studied in section 2.3)

  **Thank you for your suggestion. We changed the text accordingly:**
  *The latter two events resemble the two real rain events that are studied in Sect. 3.1.*

- L218: Is the noise that you are adding the same kind as the noise in the measurement data (white noise vs. more autocorrelated) and if not, why is it still valid to add the normally distributed white noise.
  **The noise levels are based on the measurement data. We determine these levels by averaging hourly standard deviations for all dry hours in the dataset (see sentences following after this line). Moreover, we are interested in relatively long timescales, so that the influence of the kind of noise becomes less important.**

- Section 2.3: The events start and end at the full hour. Did you also randomly adjust the temporal alignment with the full hour? It seems that this alignment could have a relevant influence when using longer aggregation times. Please comment and potentially extend your analysis in that regard.

  **See our reply to your main comment.**
- Section 3.1: I suggest to make it clear in the section title that two individual events are analysed here for better illustration.

  **We agree and adapted the title as follows:**
  *Influence of temporal sampling on rainfall estimates for two individual events*

- L240: „Yet, this does not seem to affect the computed rainfall intensity using the min-max sampling strategy." To which plot and which exact feature of the min-max rainfall intensity does this refer? Please explain and maybe also slightly rephrase.
  **We agree and rephrased as follows:**
  *Yet, this only seems to affect the difference between minimum and maximum intensity values (Figs. 4d, l and t), but not the computed rainfall intensity using this sampling strategy (Figs. 4h, p and x).*

- Fig 6: I suggest to write „Nokia 38 GHz H" and „RAL 26 GHz H" in the legend so that this is consistent. That also makes it clearer just from the plot that Fig 7 shows something different (noiseless devices)
  **We agree, see our reply to your first main comment.**

- L276 to L278: I think this statements here should be refined. From Fig 6, I would say the instantaneous sampling is clearly performing worse for the theoretical evens with added noise. That could be stated here.

  **We agree that in the overall picture, the instantaneous sampling strategy is performing worse compared to the other strategies. We revised as follows:**

  *Generally, this shows that each of the sampling strategies is capable of producing correct rainfall estimates, especially for the shortest time intervals. For longer time intervals, in particular the instantaneous sampling strategy does not perform as well as the other two sampling strategies. Additionally, the influence of noise on rainfall intensities cannot be neglected when using CMLs to measure rainfall.*

- Fig 6: Why is there such a significantly lower R_squared for instantaneous sampling for the 26 GHz , even for 1 sec data?

  **This is caused by the relatively low signal-to-noise ratio for the 26 GHz link in comparison to the 38 GHz link. Following from the R-k relationship, the attenuation for a 26 GHz link is lower than the attenuation for a 38 GHz link at the same rainfall intensity (e.g., for our setup a 2 mm h$^{-1}$ intensity results in an attenuation of 1.2 dB for a 38 GHz device and an attenuation of 0.5 dB for a 26 GHz device). The RAL 26 GHz and 38 GHz links both have noise added for the theoretical events to the signal with a standard deviation of 0.2 dB, so that the relative noise level for the RAL 26 GHz link is higher than for the 38 GHz link. Especially for low intensity events, this causes the lowest attenuations to end up below 0 dB, so that the rainfall retrieval algorithm corrects the attenuation to 0 mm h$^{-1}$. We also add this to the text:**

  *The instantaneous sampling strategy for the 26 GHz link with noise added performs significantly worse than the other strategies and links in terms of RMSE and the $r^2$ for the low-intensity event. This is caused by the relatively low signal-to-noise ratio for the 26 GHz link in comparison to the 38 GHz link. Following from the R-k relationship, the attenuation of a 26 GHz device is lower than the attenuation of a 38 GHz device for the same rainfall intensities (e.g., for our setup a 2 mm h$^{-1}$ rainfall intensity results in an attenuation of 1.2 dB for a 38 GHz device and an attenuation of 0.5 dB for a 26 GHz device). Especially for low rainfall intensities, this causes that the added noise occasionally compensates for the rainfall attenuation, so that some of the attenuations are negative. In the rainfall retrieval algorithm these negative attenuations are corrected to 0 dB, also affecting the overall statistics.*

- Section 3.3: The title is exactly the same as in section 3.1. Please correct.

  **Our apologies for the sloppiness. We changed it as follows:**
  *Influence of temporal sampling on rainfall estimates for all events*

- Fig 8: I suggest to add a selection of the scatter plots that are in the supplementary material to Fig 8. Maybe select the plots that are needed most for the discussion in the text. Given the space an individual scatter plot needs, one can easily fit 4 or 4x3 or maybe 3x3 into one figure. Or, since I suggest to focus on 1-minute instantaneous vs. 15-minute min-max (see my main comment) maybe add these two here.

  **The plots we added in the original manuscript are the plots we deem to be essential for understanding the storyline. We do realize that the order and placement of the plots is not**

**always ideal. Therefore, we have changed their locations somewhat (see also our reply to your 3rd main comment).**

- Fig 9: If I understand correctly, the min-max results that are shown here are the ones with the optimised alpha, correct? If yes, why did you chose to show the optimised min-max results here and not the ones with the default value of alpha?

  **It is indeed not entirely clear which min-max sampling strategy is presented in this figure. We use the optimized version of this sampling strategy, because for shorter and longer time intervals than 15 minutes an optimization would be required anyway (see Figure 11), as the default values were optimized for 15-minute intervals. By optimising, we present the best possible statistics using this method and do not introduce an additional source of uncertainty into the study, which would complicate decomposing the different sources of errors and uncertainties.**

  **To clarify we use optimized values, we added the following at the end of step 6 in the Methods section (about optimization):**
  *Note that, unless specifically mentioned, when we refer to min-max sampling strategies in the text and figures, we refer to the optimized version of this sampling strategy. This way, we prevent introducing an additional source of error and uncertainty into the study.*

  **When referring to the non-optimized version, we also refer the reader to the supplementary materials:**
  *This also causes all the min-max lines in Figure 9a to be close to zero. The non-optimized min-max sampling strategy performs slightly worse than the optimized strategy, but no major differences between both occur, except for the longest time intervals (see supplementary materials).*

- L303: What is meant with „reducing influence of the power law" here? Do the authors mean the impact of the non-linearity of the power law, which is smaller for small attenuation values (from averaging) than for larger attenuation values? Please be more precise in this sentence.
  **We agree that this could be better formulated. We rephrased as follows:**
  *The first is the reduced impact of the non-linearity of the power law, which decreases as a consequence of longer averaging time intervals resulting in lower average attenuation values (step 7 in Sect. 2.2).*

- L313: It is hard to understand this sentence. After reading it multiple times I now understand that the slope of the regression, which goes thought the origin, is below 1. But please rephrase.
  **We agree and rephrased as follows:**
  *Still, for the mean sampling strategy, the overall slope of the linear regression line, which goes through the origin, is below 1, which suggests that the mean sampling has a slight tendency to underestimate the rainfall intensity*

- L324 to 326. What is the reason for the fairly good metrics of the RAL link? Is it wet antenna that is still present at the end of the events? This is not clear from the text. (Update: After reading the paragraph starting at L330 this is now clear. The structure of the text is a bit strange here, though. It is not clear to me how the content is distributed to the different

paragraphs and how the paragraphs are linked, maybe because the sentence at L330 does not make that clear. I suggest to restructure the text/paragraphs here).

**We agree that the structure of the paragraph is a bit off. We removed some text after the statement originally between L324-326, which did not really add any essential information.**

*We would have expected that the rainfall intensity measured at the end of a long interval does not provide any information on the other rainfall intensities during that interval, given the 30 min average duration of rainfall events in our data. The Nokia link behaves more as expected (Fig. 10a), as the spread is larger, also resulting in a lower $r^2$ , i.e., 0.33 versus 0.64, at a 60-min time interval for instantaneous sampling.* *.*

*Based on the individual events, it seems that the majority of the differences in performance between the devices for the instantaneous sampling strategies are related to the variations in rainfall intensity, i.e., 30 min average duration (Fig. 2), in combination with wet-antenna attenuation. This is supported by the absence of any similar differences in the theoretical events. For the shortest time intervals, the rainfall intensities, and thus measured attenuations, do not vary much in these few seconds, which results in relatively good scores………..(paragraph continues)*

- L330: What is „these difference" referring to? I checked the last sentences before, but could not find a suitable statement. Please make this clearer in the text.

  **It is indeed not clear, we rephrased as follows:**

  *Based on the individual events, it seems that the majority of the differences in performance between the devices for the instantaneous sampling strategies are related to the variations in rainfall intensity, i.e., 30 min average duration (Fig. 2), in combination with wet-antenna attenuation*

- L341: With „the fluctuations around the mean" do you mean fluctuations caused by noise, or the rain-induced changes of the attenuation? Maybe it would be best to not use the term „fluctuation" here and in the next two sentences.

  **We refer to the minimum and maximum attenuations with "fluctuations", though that is not clear from the text. We rephrased as follows:**

  *For the shortest time intervals, this points to a roughly symmetrical distribution of the minimum and maximum attenuations around the mean. For the longer time intervals, this suggests a positively skewed distribution of the attenuations with relatively many outliers to the maximum attenuation. For the non-optimized min-max sampling…*

- L356 to L365: If I understand correctly this describes a shortcoming of the way the maximum power level is treated, in combination with how the baseline is set, resulting in overestimation during low attenuation rain events. This seems to also apply to how CML processing is done e.g. in RAINLINK. Hence, this is a very relevant detail. It is, however, hard to understand the explanation. Having a plot that support the explanation would be good. Rephrasing the text might still be required.

**We agree that this is an important detail. We tried to use figures 4 & 12 as supporting figures for this, however, we realize that this was not clearly formulated. We rephrased the entire paragraph as follows:**

*Moreover, for low rainfall intensities, especially below 2 mm h$^{-1}$, the rainfall retrieval algorithm for the min-max sampling strategy overestimates rainfall intensities (Fig. 12). Reasons for this seem to be twofold. Firstly, the minimum attenuation is set to 0 dB km$^{-1}$, due to the maximum received power (resulting in minimum attenuation) being higher than the baseline power level. Part of this is caused by the assumption that the median of the average of the minimum and maximum received power levels represents the baseline is not always valid, due to a skewness towards minimum power levels. For example, between 9:00 and 11:00 in Fig. 4t for the hourly time intervals, the maximum attenuations are constant, while the minimum attenuation changes between the intervals 9:00-10:00 and 10:00-11:00. Still this results in a constant rainfall intensity (Fig. 4x). Secondly, the minimum received power level (resulting in the maximum attenuation) is nearly always significantly less than the baseline power level. This means that for low rainfall intensities, especially around and below 1 mm h$^{-1}$, the minimum attenuation is corrected (i.e., set to 0 dB km$^{-1}$), while the maximum attenuation is treated as is, preventing the min-max retrieval algorithm to work the way it was designed. In the rainfall intensity computation, both attenuations are treated in a similar manner, giving too much weight to the maximum attenuation, causing an overestimation of the rainfall intensity. For slightly higher precipitation intensities (1-2 mm h$^{-1}$), the maximum power levels are still close to the baseline, so that barely any increase in rainfall intensity in the min-max sampling is computed, while the 20 Hz rainfall intensity estimates do increase, causing the bend seen in Fig 12. For even higher rainfall intensities (>2 mm h$^{-1}$), both the minimum and maximum received power levels are lower than the baseline, so that the min-max sampling method can be used the way it was designed.*

- L381: „This shows that the performance of the sampling strategy and rainfall retrieval algorithm is largely dependent on the wet-antenna attenuation and differences in baseline power levels". This sounds as if the significant temporal undersampling of instantaneous sampling with long intervals has a smaller effect than wet antenna and baseline power levels. Is this really the case?
  **We mean here that the sensitivity of the performance of the sampling strategy is also affected by the wet-antenna attenuation and differences in baseline power levels. We recognise that the current formulation is too strong. We adapted as follows:**
  This shows that the wet-antenna attenuation and differences in baseline power levels can play a relatively important role in the performance of the sampling strategy, next to the sensitivity to sampling interval and method.

- General statement on Section 3.3: This section is too long and covers too many different details. I suggest to either add sub-subsection (maybe without numbering) or to split the content into more subsections. Since I also suggest (see my main comments) to add an analysis that focuses on the comparison of 15-minute-min-max with 1-minute-instantaneous (maybe in addition 30sec or 1sec), some content could be redistributed.
  **See our reply to your main comment**

- L421: I do not see why it should be „surprising" that the instantaneous sampling performs well for shortest time intervals. If the sampling is done fast enough to avoid undersampling of the

rain-induced signal changes, it will capture the relevant dynamics very well, albeit being more affected by instrument noise than e.g. mean sampling.

**We agree that this is not necessarily surprising. We removed "surprisingly" from the sentence.**

- L433: „Our results are in line with Pudashine et al. (2021)…". From Fig. 9 I conclude that mean sampling is clearly better than min-max at 15-minute sampling for correlation and RMSE (MBE is not a fair comparison because of optimised min-max). Hence, I do not see how your results are in line with their results.

**We agree that this statement is not entirely correct. However, the differences found between these sampling strategies are not very large, especially in comparison to the instantaneous sampling strategy. Additionally, it should be noted that different reference data were used. We rephrased as follows:**

*Our results are not fully in line with Pudashine et al. (2021), who show that the min-max sampling strategy slightly outperforms the mean sampling strategy, though it should be noted that their study uses gauge-adjusted radar data as reference, which makes an objective comparison difficult.*

- L435: I do not find any analysis of the effect of quantisation in this manuscript. How can your results be in line with Ostrometzky et al. (2017)? Please elaborate also in the manuscript or remove this statement.

**We agree that this is not the best place to refer to Ostrometzky et al. (2017), as we mostly mean this comment to emphasise the importance of quantization. Therefore, we moved this sentence to the final paragraph in the discussion:**

*Leijnse et al. (2008) demonstrate that power quantization can have a significant effect on the estimated rainfall intensities when using CML networks, especially for low rainfall intensities. Ostrometzky et al. (2017) show that min-max sampling combined with the quantization effect can lead to significant biases for rainfall retrieval. Chwala and Kunstmann (2019) show that the quantization effect limits the minimal detectable rainfall intensity.*

- L445: „…likely caused by an internal filter". This filter was mentioned already before. But how do you know that there is such a filter. Maybe the Nokia CML's hardware is just more sophisticated with better low-noise amplifiers and/or better shielding from external disturbances?

**We agree that we don't exactly know what causes the differences between the radio links. Therefore, we rephrased all references to the filter in the text to different hardware, as a more generic term.**

*L106-110: The employed frequencies for the Nokia and RAL 38 GHz links are close, hence exhibit similar electromagnetic characteristics, but do not interfere with each other. However, these devices were found to give a different response, likely due to the internal hardware in the Nokia link being designed differently, reducing the high-frequency fluctuations in the signal, while the RAL link has a different antenna cover than the Nokia link, which affects the distribution of water remnants on the cover (see van Leth et al., 2018a).*

*L242-243: These high-frequency fluctuations in the signal are roughly reduced by 0.5 dB, which is likely caused by the different internal electronics in the Nokia link.*

*L375-376: Overall, this indicates that a reduced duration of wet-antenna attenuation and hardware reducing the signal fluctuations can significantly reduce the influence of the selected temporal sampling strategy.*

*L447-451: An additional difference between these devices is the reduced signal fluctuation in the Nokia link, likely caused by the different hardware employed in the Nokia link. However, these differences do not have an influence of the same order of magnitude on the raw signal. Where the hardware causes the fluctuations to reduce roughly by 0.5 dB, the additional wet-antenna attenuation for the RAL link is roughly 2 dB higher. Therefore, we attribute the largest differences between the Nokia and RAL 38 GHz links to the difference in wet-antenna attenuation.*

*L555-556: This device mostly differs from the other two devices, the RAL 38 and 26 GHz links, in terms of reduced magnitude and duration of wet-antenna attenuation and is designed with hardware that reduces signal fluctuations.*

- L458: „…we do not expect this mismatch in timescales to have a significant effect". Since shorter event will have an impact already for sampling with shorter intervals, this could have an effect on your results. You do not sample with spacing longer than 60 minutes. Hence, your theoretical rain events are longer than your longest sampling interval. Please comment and adjust accordingly.
  **We tried to refer in this sentence to the effect of wet-antenna attenuation and different baseline power levels on the statistics and the accompanying conclusions. We agree that the current phrasing suggests differently. Indeed in general these different timescales with "undersampling" of a rainfall event can significantly affect our results. We rephrased as follows:**
  *….but was chosen to resemble the individual events. However, we do not expect this mismatch in timescales to have a major effect on the differences between the theoretical and actual data, i.e., the influence of wet-antenna attenuation and different baseline power levels. Overall,…*

- L460: „Additionally, it would create more need to adjust for the instrumental bias." I do not understand what this sentence means.
  **We refer to the instrumental bias of the microwave links itself and not the biases that occur during the processing of the data. We added the following:**
  *Additionally, it would create more need to adjust for the instrumental bias of microwave links, for example as a consequence of different antenna covers or temperature dependence (e.g., van Leth et al., 2018a).*

- L462: General comment on this paragraph. The most comprehensive and very recent comparison of different wet antenna methods is given by Pastorek et al. (2021) https://doi.org/10.1109/TGRS.2021.3110004 . This paper should be added here.
  **We agree that this paper gives a good overview of the various corrections. We added as follows:**
  *Graf et al. (2020) found that for a telecom network in Germany a correction based on rainfall intensity (based on Leijnse et al., 2008) outperformed a method based on the time (based on Schleiss et al., 2013) during and after a precipitation event. Similarly, Pastorek et al. (2022) compared multiple wet-antenna attenuation corrections and also concluded that corrections based on rainfall intensity outperformed other methods. Additionally, they found that these*

*corrections can be applied to intensities obtained from sub-links with various frequencies and path lengths, and thus can also be applicable to other networks with similar antenna characteristics.*

- L470: „…a correction based on rainfall intensity outperformed a method based on the time…" Graf et al (2020) used the methods that are described in this paragraph, the one from Schleiss et al (2013) and the one from Leijnse et al (2008), but with adjusted parameters.
  **We added the following:**
  *Subsequently, Overeem et al. (2016a) applied this value as a constant for correcting attenuation due to wet antennas for each microwave link. Leijnse et al. (2008) propose to use a more physics-based model to compute the wet-antenna attenuation, which uses signal frequency, antenna cover properties and rainfall intensities and seems especially useful for shorter time intervals. Graf et al. (2020) found that for a telecom network in Germany a correction based on rainfall intensity (based on Leijnse et al., 2008) outperformed a method based on the time (based on Schleiss et al., 2013) during and after a precipitation event. Similarly, Pastorek et al. (2022)….. (paragraph continues)*

- L483: „…when using the same device as reference data." I do not understand what is meant here.
  **We agree that this can be more clearly formulated. We rephrased as follows:**
  *…especially wet-antenna attenuation, baseline variation and the non-linear effect of the power law, affect the performance of the rainfall estimates, even when using the same microwave link as reference.*

- L491: „In general, our efforts allow future studies to focus on estimating the uncertainty of their observed rainfall intensities using microwave links and uncover the instrumental bias of these links". I do not understand how this should be done. Maybe it is described somehow in the sentences before, but this is not clear (to me). Please rephrase, potentially also the sentences before, or add a more detailed explanation.
  **This is indeed a too strong formulation. We rephrased as follows:**
  *In general, our efforts allow future studies to estimate the uncertainty of their observed rainfall intensities as a consequence of the chosen sampling strategy and potentially uncover the instrumental bias of these links.*

- L494: The fact that CML lengths can be very different in a real network is mentioned here, but it is not discussed in the text below. Since increasing path length will decrease the variability of the rain-induced path-attenuation, it might have an effect on the time scales at which a sampling strategy starts to show significant decrease of performance. The study from Leijnse et al (2008), albeit using a smaller number of sampling variants, includes CML path length. I suggest to discuss the effect of path length and its interplay with sampling strategy, potentially using the results from Leijnse et al (2008) to estimate an extrapolation of your results to different path lengths.
  **This indeed is an important point to mention. We added as follows:**
  *For an increase in link length, we expect a reduced sensitivity of the sampling strategies to increasing time intervals, while also the differences between sampling strategies decrease. As suggested by Leijnse et al. (2008), this is caused by the increase in characteristic timescales of*

*the path-averaged rainfall intensities when increasing link lengths. Similarly, Berne and Uijlenhoet (2007) showed for longer link lengths a decrease in uncertainty in rainfall estimates and, moreover, a reduction in sensitivity to sampling effects.*

- L567: „…independent of the selected sampling strategy". But for sampling on short intervals, wet antenna cannot have an effect in your analysis because the drying periods are considered dry based on disdrometer data. Hence, I find this statement a bit confusing. Of course, wet antenna has a „significant influence" on the „rainfall estimates" when compared to reference data, but this is not done in this manuscript. Please rephrase.

  **Here, we refer to the fact that wet-antenna attenuation does have an significant effect on all sampling strategies during rain events, based on our comparison of the actual events to the theoretical events (see preceding sentences in manuscript). This is currently not reflected in the text, therefore we rephrased as follows:**
  This illustrates the significant influence wet-antenna attenuation during rain events can have on the rainfall estimates, for all sampling strategies.

Technical corrections

- L35: I think it would be easier to read with „radars measure…" instead of „radar measures".
  **We agree. We changed accordingly:**
  *However, radars measure higher up in the atmosphere and indirectly retrieves rainfall introducing uncertainty….*

- L42: „… CMLs are deployed, of which…" I suggest to find a better formulation here. This is hard to understand.
  **We rephrased as follows:**
  *These networks consist of commercial microwave links (CMLs), the rain-induced attenuation of the electromagnetic signal of which can be used to compute rainfall intensities.*

- L45: Not sure but a comma might be required after the „thus"
  **We agree, it indeed seems better to add the comma there. We changed as follows:**
  *Thus, as a major advantage, the infrastructure required to spatially measure rainfall with these CMLs already exists.*

- L114: Maybe better write „less prone to" instead of „prone to less"
  **We agree, we changed accordingly:**
  *The RAL 26 GHz link is less prone to wet-antenna attenuation than the RAL 38 GHz link.*

- L260: not sure, but maybe better to write „from the instruments"
  **We agree. We changed accordingly:**
  *This allows us to determine which part of the uncertainties originates from the rainfall retrieval algorithm and which from the instruments.*

- L327: add „is" after „We except that this"
  **We agree. We changed accordingly:**

We expect that this is caused by the large sample size, which reduces these uncertainties.

- L458: Better write „the two individual events studied in section 3.1" to be clearer
  **We agree. We changed accordingly:**
  *but was chosen to resemble the two individual events studied in Sect. 3.1.*

- L528: Remove one „with"
  **We indeed wrote a "with" too many. We changed as follows:**
  *We compared the mean, instantaneous and min-max sampling strategies and various time intervals ranging from 1 s to 60 min with with 20 Hz rainfall estimates of the same device, allowing us to exclude the direct instrumental bias in this comparison.*

---

## Author Comment (AC3)

**Dear Referee,**

**We would like to thank you for taking the time to review our paper and for your suggestions, which helped to improve the quality of the manuscript. We reply to your comments below. Our response to the comments appears in bold and revised text as** *italic***.**

- P2, L52: "complimentary" -> complementary
  **We agree and changed as follows:**
  *van het Schip et al. (2017) showed the complementary potential of CML and satellite data by determining wet and dry periods using the satellite data, while Hoedjes et al. (2014) proposed to use this for a conceptual flash flood early warning system in Kenya.*

- P3, L69. Papua New Guinea is actually in Oceania. Do you actually mean Papua New Guinea or some other country (Guinea, Equatorial Guinea, Guinea Bissau) that is actually in Africa?
  **We made a mistake and indeed mean Papua New Guinea. We changed this as follows:**
  *Since then, studies have been performed in Europe (Czech Republic, France, Germany, Italy, Luxembourg, Sweden and Switzerland), Africa (Burkina Faso, Kenya and Nigeria), South America (Brazil), Asia (Lebanon, Pakistan and Sri Lanka) and Oceania (Australia and Papua New Guinea) (see Chwala and Kunstmann, 2019, for a partial overview)*

- P3, L77-78. Consider rephrasing as "minimum and maximum values (and occasionally mean and/or instantaneous values) are most commonly measured with a temporal resolution of 15 minutes"
  **We agree that this is suggestion is a better formulated sentence:**
  *Moreover, not all mobile network operators store the same variables describing the link signal in their network management system. Minimum and maximum values (and occasionally mean and/or instantaneous values) are most commonly measured with a temporal resolution of 15 minutes. Additionally, ….*

- P8. L203-204. "This makes that... are different..". Consider rephrasing as "This causes the rain intensities... to be different..."
  **We agree and changed accordingly:**
  *This causes the rain intensities obtained by using averaged attenuation to be different compared to the averaged rainfall intensities.*

- P11, L327, "We expect that this caused..." -> "We expect that this is caused..."
  **We agree, the word "is" is missing in this sentence. We have added it to the new version:**
  *We expect that this is caused by the large sample size, which reduces these uncertainties.*

---

## Author Response (AR2)

**Dear Referee,**

**We would like to thank you for taking the time to review our paper and for all your constructive suggestions, which definitely helped to improve the quality of the manuscript. We reply to your comments below. Our response to the comments appears in bold and revised text as** *italic*.

General comments:

- 1. I appreciate that the authors have added markers to certain figures to highlight 1-minute instantaneous and 15-min min-max sampling. I do find the markers a bit too large, though. Maybe there was a reason to use the large crosses as markers. But smaller makers, maybe with a thiner edge and maybe some transparency could mitigate the overplotting that is unavoidable in the figures. Or maybe you could give the individual markers a small horizontal offset. Or maybe use three different maker types?

  **We agree that it was somewhat hard to see where all markers were. Therefore, we have changed the markers to transparent markers with a different shape per device. Together with your third comment, the figures are now as follows:**

**Fig. 6:**

[Figure]

**Fig 7:**

[Figure]

- 2. Reading through the paper and looking at the (updated) plots I am again automatically concentrating mostly on a comparison between 1-minute instantaneous and 15-minute min-max. I understand that the authors want to provide, as they wrote in their response letter, a systematic overview of the consequences of different sampling strategies. However, as I already stated in my last review, there is significant importance to the comparison between 1-minutes instantaneous and 15-minutes min-max. There is still not a dedicated discussion on this, even though some figures have now been updated to mark these two different sampling strategies. Since I was again dedicatedly looking for this information, I have two suggestions:
2a: It would be beneficial to have a short (sub-)subsection in section 3 or 4 to discuss the comparison of 1-minute instantaneous and 15-minute min-max.

**We added the following paragraph to Section 4.1:**

*When comparing the conventional 15-minute min-max and 1-minute instantaneous sampling strategies, the overall statistics are similar. For the theoretical low-intensity event with sliding window for a 38 GHz noiseless link (Fig. 8), the 15-minute min-max sampling strategy shows a reduced performance in comparison to the 1-minute instantaneous sampling strategy. For the high-intensity event, the performances of the sampling strategies are on average more comparable. Overall, this could imply that in regions where low rainfall intensities are more prevalent than high intensities, it could be beneficial to use a 1-minute instantaneous sampling strategy instead of the 15-minute sampling strategy. For the actual data (Fig. 10), the $r^2$ and the slope for the 15-minute min-max sampling strategy are on average for all devices slightly lower (i.e., further from one) than for the 1-minute instantaneous sampling strategy. However, these differences are not of the same order of magnitude as differences with other time intervals. A reduction in sampling time interval for both sampling strategies (namely instantaneous versus min-max) would result in a larger increase in performance than the difference in performance between these sampling strategies, especially for the 15-minute min-max sampling strategy.*

2b. It would be beneficial to add a version of Fig 5 (and also Fig 4) to the supplementary material that only shows 20 Hz, 1-minute instantaneous and 15-minute min-max, maybe also 1-second instantaneous because some CML data is available at 10-second resolution and higher resolution data might become available the future. If you use separate columns with one line each, you could add the 20 Hz data always in the background.

**Thanks for the suggestion. We have added similar figures as Figs. 4 & 5 in the supplementary materials, but with 1-minute and 15-minute lines (Figs. S5 & S6). This allows for an easy comparison with Figs. 4 & 5.**

**Example Figure S6:**

[Figure]

- 3. The new results in Fig 8 are interesting and well described. However, I found it complicated to visually compare the results to Fig 7 and Fig 6 to understand how the temporal shift affects performance compared to adding noise. Since the colors are the same but with different meaning in Fig 8, and since the x axis is linear instead of log in Fig 8, it requires some effort to detect differences. I wonder if this could be improved, but I do not have a solution. Maybe an easy fix could be to add data from Fig 6 or Fig 7 with a very light color in the background. But this could also make the plot too busy.

  **We agree that is hard to compare these figures with each other. We have switched the colour and line styles in Figs. 6,7 & 10, so that the colours are consistent with Fig. 8. See our reply to your first comment for the new figures.**

Specific comments:

- Fig 6: The updated information in the text clearly explains why R_2 is so low for the low intensity event. But why is this not happening in Fig 10? I assume that most rainfall in the real dataset comes at low intensities, hence R_2 is most likely dominated by these low rainfall intensities. But at 1-minute instantaneous sampling there is only a very small difference between RAL 26 and RAL 38 in Fig 10, even though RAL 26 has a high noise level relative to the rain-induced attenuation (as you describe in your updated text related to Fig 6). It would be good to explain this behavior because, from Fig 6, one concludes that the (relative) noisiness of RAL 26 has a significant impact, while for the statistics from the real dataset in Fig 10 one would not draw that conclusion. Maybe wet antenna increases the attenuation in the real data so that low attenuation during low rainfall is less often smaller than zero (which happens for the theoretical events due to added noise)?

  **Indeed, a similar behaviour would be expected in the real dataset. We attribute this as you suggest to the wet-antenna attenuation, which increases the attenuation in the real dataset. We have added as follows:**

  *It should be noted that differences in baseline power levels between sampling strategies and wet-antenna attenuation are not included in these theoretical events, while in Section 3.1 these clearly affected the rainfall intensity estimates.* *For example, wet-antenna attenuation makes the previously described behaviour of the instantaneous sampling strategy for the RAL 26 GHz link less likely to occur in the actual data, because the lowest attenuation levels will increase as a consequence of wet antennas.* *Differences in baseline power levels are only slightly reflected in the theoretical events as caused by the added noise, which might slightly affect the median signal intensity for the computation of the baseline power levels.*

- L284: „…instantaneous and min-max sampling strategies seem to be more prone to errors in retrieving high rainfall intensities…" Some lines below you write that instantaneous sampling is good in capturing the peaks intensities. This is also what I would get from Fig 5, of course not for 60-minute aggregation. But at 5-minute sampling the height of the second peak is, surprisingly, best captured with instantaneous sampling. Maybe this is a general tendency because the mean can easily get biased towards underestimation, as you write elsewhere. But I am not sure if this is the reason here.

  **Based on Figs. 4 & 5, we tried to explain here that for the shortest time intervals (e.g., 1-second intervals) the instantaneous sampling strategy is able to capture the peaks (because of the short time intervals), while for longer time intervals this is not necessarily the case (as one would also expect). Indeed the 5-minute sampling in Fig. 5 quite nicely captures the peak intensity, because the sampled intensity is representative for the entire time interval. We did not mean to refer here to the subsequent sections where the underestimation of the mean sampling strategy is addressed. We added some references to Fig. 5 and added as follows:**

*Generally, the instantaneous and min-max sampling strategies seem to be more prone to errors in retrieving high rainfall intensities. For the event on 21 June (Fig. 5), the mean sampling strategy results, as expected, in decent estimates of the evolution of the rainfall event on average, both in timing and intensities (though the peak rainfall intensities are obviously averaged out). The instantaneous sampling strategy seems to be most sensitive to an increase in the length of the time interval, because its performance depends on the representativeness of a single measurement for the whole time interval. The timing and intensity of the estimated peak rainfall intensity for the shortest time intervals are often relatively good, due to the large signal intensity fluctuations as a consequence of variations in rainfall intensity in comparison to the instrument noise, while for longer time intervals the sensitivity of the performance to the representativeness of a single measurement for the whole time interval heavily increases (e.g., the 5-minute instantaneous sampling strategy in Fig. 5 captures the peak intensity at 12:00 better than the mean and min-max sampling strategies). For the min-max sampling strategy, the timing of the peak intensity is generally well-captured, but the estimated peak rainfall intensity can be inaccurate. Additionally, for this specific case, this method strongly overestimates the rainfall sum for the 60-minute interval, due to the peak taking place around the full hour, so that two subsequent intervals cover this peak.*

- L289: What is meant with „signal-to-noise" ratio here in the context of comparing instantaneous with the other sampling methods?

  **We refer here to the relatively large fluctuations in signal intensity as a consequence of changes in rainfall intensity in comparison to the instrument noise. To clarify, we added as follows:**

  *The timing and intensity of the estimated peak rainfall intensity for short time intervals are often relatively good, due to the large signal intensity fluctuations as a consequence of variations in rainfall intensity in comparison to the instrument noise, while …..*

- L290: Which „sensitivity" increases with larger time intervals?

  **We refer here to the sensitivity of the representativeness of a single measurement for the whole time interval. We realize that this could be phrased differently. We changed as follows:**
  *….longer time intervals the sensitivity of the performance to the representativeness of a single measurement for the whole time interval heavily increases….*

- L587-L591: What about shorter link length? Of course, one expects the opposite of what happens for longer link lengths (which is described in the new text). But given that it has been reported that very short instantaneously sampled CMLs (with lengths of hundreds of meters or even only tens of meters) are prone to overestimation of rain rates, can you make a statement on the question if instantaneous sampling is a potential reason for this overestimation and not (only) the wet antenna estimation (which is undoubtedly a large

source of error for very short and thus insensitive CMLs)? If I assume that the positive MBE of instantaneous sampling around 5-minutes in Fig 10 is shifted to the left (maybe to 30-s or below) for very short CMLs, that could be the case.

**Thank you for this suggestion. This is indeed a connection that we had not made yet. We added the following lines at the end of this paragraph:**

*.... For shorter link lengths, an opposite behaviour is expected. This could, for example, imply that part of the overestimations found for short 38 GHz links are not only a consequence of wet-antenna attenuation (Fencl et al., 2019), which does play a significant role, but also of sampling strategy.*